# SHP2 as a primordial epigenetic enzyme expunges histone H3 pTyr-54 to amend androgen receptor homeostasis

Surbhi Chouhan[1,2,14], Dhivya Sridaran [1,2,14], Cody Weimholt[3], Jingqin Luo [4,5], Tiandao Li [6], Myles C. Hodgson[7], Luana N. Santos[7], Samantha Le Sommer[7], Bin Fang [8], John M. Koomen [8], Markus Seeliger [9], Cheng-Kui Qu [10], Armelle Yart[11], Maria I. Kontaridis [7,12,13], Kiran Mahajan[1,2] & Nupam P. Mahajan [1,2,5] ✉

Mutations that decrease or increase the activity of the tyrosine phosphatase, SHP2 (encoded by *PTPN11*), promotes developmental disorders and several malignancies by varying phosphatase activity. We uncovered that SHP2 is a distinct class of an epigenetic enzyme; upon phosphorylation by the kinase ACK1/TNK2, pSHP2 was escorted by androgen receptor (AR) to chromatin, erasing hitherto unidentified pY54-H3 (phosphorylation of histones H3 at Tyr54) epigenetic marks to trigger a transcriptional program of AR. Noonan Syndrome with Multiple Lentigines (NSML) patients, SHP2 knock-in mice, and ACK1 knockout mice presented dramatic increase in pY54-H3, leading to loss of AR transcriptome. In contrast, prostate tumors with high pSHP2 and pACK1 activity exhibited progressive downregulation of pY54-H3 levels and higher AR expression that correlated with disease severity. Overall, pSHP2/pY54-H3 signaling acts as a sentinel of AR homeostasis, explaining not only growth retardation, genital abnormalities and infertility among NSML patients, but also significant AR upregulation in prostate cancer patients.

The catalytic activity of the protein tyrosine phosphatase (PTP) superfamily of enzymes is often regulated by tyrosine kinases and has been shown to play a key role in various physiological processes[1–3]. Src homology-2 domain-containing PTP2 (SHP2) is a ubiquitous non-receptor PTP encoded by the *PTPN11* gene, composed of two SH2 domains at the N-terminus, followed by a PTP domain and two tyrosine residues Tyr542 and Tyr580, at the C-terminus targeted for phosphorylation[4–6]. SHP2 is self-inhibited by interaction of N-terminal SH2 domain with the PTP domain[7,8]. Activated SHP2 plays a pivotal role in modulating cytosolic signaling cascades including RAS/ERK, MAPK/

[1]Department of Surgery, Washington University in St Louis, St Louis, MO 63110, USA. [2]6601, Cancer Research Building, Washington University in St Louis, St Louis, MO 63110, USA. [3]Department of Pathology and Immunology, Washington University in St Louis, St Louis, MO 63110, USA. [4]Division of Public Health Sciences, Washington University in St Louis, St Louis, MO 63110, USA. [5]Siteman Cancer Center, Washington University in St Louis, St Louis, MO 63110, USA. [6]Bioinformatics Research Core, Center of Regenerative Medicine, Washington University in St Louis, St Louis, MO 63110, USA. [7]Department of Biomedical Research and Translational Medicine, Masonic Medical Research Institute, 2150 Bleecker St, Utica, NY 13501, USA. [8]Moffitt Cancer Center, SRB3, 12902 Magnolia Drive, Tampa, FL 33612, USA. [9]Department of Pharmacological Sciences, Stony Brook University Medical School, BST 7-120, Stony Brook, NY 11794-8651, USA. [10]Department of Pediatrics, Aflac Cancer and Blood Disorders Center, Winship Cancer Institute, Children's Healthcare of Atlanta, Emory University School of Medicine, Atlanta, GA 30322, USA. [11]UMR 1301-Inserm 5070-CNRS EFS Univ. P. Sabatier, 4bis Ave Hubert Curien, 31100 Toulouse, France. [12]Department of Biological Chemistry and Molecular Pharmacology, Harvard Medical School, Boston, MA, USA. [13]Department of Medicine, Division of Cardiology, Beth Israel Deaconess Medical Center, Harvard Medical School, Boston, MA, USA. [14]These authors contributed equally: Surbhi Chouhan, Dhivya Sridaran. ✉e-mail: nupam@wustl.edu

ERK, PI3K/AKT, and JAK/STAT pathways[9,10]. In addition, SHP2 also has other targets, e.g. YAP dephosphorylation at Y357 in the setting of RAS/RAF mutations, which may have role in chemoresistance in cholangiocarcinomas[11]. The clinical relevance of SHP2 signaling became apparent with the discovery of mutations that either inactivate or activate its phosphatase activity in human disorders. Catalytically activating heterozygous germline mutations in *PTPN11* are associated with half of the cases of Noonan Syndrome (NS) and are also seen in blood cancers e.g. juvenile myelomonocytic leukemia (JMML), myelodysplastic syndromes and acute myeloid leukemia[12–16]. Upregulation of SHP2 expression has also been reported in solid cancers, including in breast, pancreatic, non–small cell lung (NSCLC), and head and neck cancer[6,17–21]. In contrast, germline heterozygous mutations in *PTPN11* that decreases its phosphatase activity are associated with 90% of the developmental disorder, Noonan Syndrome with Multiple Lentigines (NSML), earlier known as LEOPARD Syndrome[22]. Recurrent mutations in *PTPN11* at Y279C, T468M and Q510E have been identified in the majority of NSML patients[23–25]. NSML is inherited as an autosomal dominant trait and its clinical manifestations includes lentigines, cardiac abnormalities, facial dysmorphism, retardation of growth, and abnormalities of the genitals such as undescended testicles[22,26].

ACK1, also known as TNK2, is a non-receptor tyrosine kinase that has emerged to be a versatile oncogenic kinase activated by point mutations, gene amplifications, or ligand-dependent activation of receptor tyrosine kinases, including EGFR, PDGFR, HER2 and IR[27–29]. ACK1 activation has been reported in prostate, breast, lung, and gastrointestinal cancers[27–37]. ACK1 activation was also observed in chronic neutrophilic leukemia (CNL) and atypical chronic myeloid leukemia (CML) with truncation mutations in CSF3R[38]. White blood cells of juvenile myelomonocytic leukemia (JMML) patient with *PTPN11* activating mutation exhibited significantly reduced viability in the presence of siRNAs targeting ACK1, indicating ACK1's role in optimal SHP2 activation[39].

Recently, the liquid-liquid phase separation (LLPS) behavior of SHP2 mutants, which stimulated SHP2 activity, has been implicated as a gain-of-function mechanism[40]. Open conformation SHP2 mutants forming discrete puncta exclusively in cytosol revealed two major caveats for tackling the functionality of this complex enzyme; first, loss-of-function mutants of SHP2 may activate other signaling pathway that is distinct from the gain of function mutants. Second, loss-of-function SHP2 mutants may exert their effect via influencing the nuclear function. However, neither the signaling outcomes of impaired nuclear phosphatase activity causing human disorder is known, nor are the nuclear targets. In this study, we report discovery of SHP2 as a chromatin modifying enzyme that specifically erase pY54-H3 (phosphorylation of histones H3 at Tyr54) epigenetic marks. Further, we demonstrate that this chromatin-altering property of SHP2 drives a diverse spectrum of disorders such as prostate cancer and NSML by regulating androgen receptor (AR) transcriptome.

## Results

### ACK1 regulates SHP2 mediated H3 Tyr54-dephosphorylation activity

Histone Tyr-phosphorylation is an epigenetic mark sparsely studied and poorly understood[41]. Earlier, we uncovered phosphorylations of histone H2B (Tyr37), and H4 (Tyr88)[28,42], and characterization of these two events indicated the presence of an additional epigenetic mark, Tyr54 in histone H3, a site that is evolutionary conserved, including in lower eukaryotes such as yeast (Supplementary Fig. S1a). Based on these studies and phosphoproteomic characterizations of multiple tissues[43], we generated antibodies that specifically recognized Tyr54-phosphorylation at Histone H3 (pY54-H3) (Supplementary Fig. S1b–d). Briefly, we assessed the reactivity of the pY54-H3 antibodies to histone H3 derived peptide (spanning 49-63 aa) and the same peptide with Y54 residue phosphorylated. The pY54-H3 antibodies recognized the H3

phospho-peptide but not the unphosphorylated H3 peptide or the non-specific phosphopeptide derived from histone H2B with Tyr37 phosphorylated (Supplementary Fig. S1b). Further, competition of pY54-H3 antibodies with H3 Y54-phospho-peptide resulted in complete loss of recognition, however, peptides derived from H2B with Tyr37 phosphorylated did not affect pY54-H3 antibody binding (Supplementary Fig. S1c). Moreover, pY54-H3 antibodies did not cross react with peptides derived from histones H2B, H3 and H4 (Supplementary Fig. S1d). Taken together with data shown in Fig. 1a, b and e (described below), these data indicate that the antibodies are selective for Tyr54-phosphorylated H3.

To determine the tyrosine phosphatase and kinase responsible for modifying H3 at Y54, we assessed multiple kinases, including receptor tyrosine kinases such as EGFR, HER2, PDGFR, FGFR, IR, nonreceptor tyrosine kinases, including SRC, ABL and ACK1, and phosphatase SHP2. Interestingly, although many of these Tyr-kinases could phosphorylate H3 at Y54, both ACK1 and SHP2 expression resulted in almost complete loss of pY54-H3 (Supplementary Fig. S1e). ACK1 activation, seen as its autophophorylation at Tyr284 site, is well studied in prostate cancer patients[28,33,37], and is also reflected in multiple prostate cancer derived cell lines. All the prostate cancer cell lines with high ACK1 kinase activity and pY580-SHP2 levels exhibited negligible pY54-H3 expression. In contrast, RWPE-1 normal prostate cells, that have low pACK1/pSHP2, exhibited robust pY54-H3 expression (Supplementary Fig. S1f). ACK1 phosphorylates SHP2 leading to activation of its Tyr-phosphatase activity, which subsequently dephosphorylated phospho-ACK1 in a negative feedback loop[39], raising the possibility that pACK1/pSHP2 signaling modulates pY54-H3 levels.

To further understand the role of ACK1-SHP2 signaling in the regulation of pY54-H3, HEK293T cells were transfected with SHP2 and ACK1 or kinase dead ACK1 (kdACK1; K158R mutation) expressing constructs[35]. ACK1 overexpression led to SHP2-phosphorylation, thereby causing complete ablation of pY54-H3 (Fig. 1a, top 3 panels). In contrast, kdACK1 was unable to phosphorylate SHP2, leading to build-up of pY54-H3 (Fig. 1a). Further, a mutant construct of H3, Y54F was generated, which was co-expressed with ACK1 or SHP2. As expected, H3 Y54F mutant exhibited complete loss of phosphorylation when probed with pY54-H3 antibodies (Fig. 1b). Treatment with ACK1 inhibitor, (*R*)-**9b**[28] or SHP2 inhibitor, SHP099[44], significantly elevated pY54-H3 levels in VCaP and C4-2B cells (Fig. 1c).

To determine whether pY54-H3 is directly dephosphorylated by SHP2, FLAG-tagged SHP2 was purified from HEK cells (Supplementary Fig. S1g, h), which was then incubated with the H3 derived peptide (spanning 49-63 aa) with Tyr54 phosphorylated. It was followed by spotting and immunoblotting with pY54-H3 antibodies. SHP2 erased Y54-phosphorylation (Supplementary Fig. S1i). As a control, purified SHP2 protein was incubated with phosphopeptides corresponding to two other known histone phosphorylations, pTyr37 in H2B[42], and pTyr88 in H4[28]. The phospho-peptides corresponding to these two modified residues were unaffected by purified SHP2 (Supplementary Fig. S1j, k). Further, ac130-H2A[45] and pY363-AR peptides[33] were used as additional controls, which were also not targeted by SHP2 (Supplementary Fig. S1l, m). In addition, purified nucleosomes were incubated with purified ACK1[28] or purified SHP2, followed by immunoblotting. ACK1 enhanced pY54-H3 in nucleosomes (Supplementary Fig. S1n, lane 2). Further, VCaP cells were transfected with SHP2 substrate trapping mutant, C459S/D425A[46]. The SHP2 substrate trapping mutant, C459S/D425A exhibited significant increase in its binding to pY54-H3 (Supplementary Fig. S1o). Furthermore, the SHP2 knockdown by siRNA reduced SHP2/H3 complex formation (Supplementary Fig. S1p). Overall, these data indicate direct dephosphorylation of pY54-H3 by SHP2.

Increasing concentration of (*R*)-**9b** or SHP099 resulted in pY54-H3 elevation in concentration dependent manner (Supplementary Fig. S2e). Further, HEK293T cells were co-transfected with FLAG-tagged SHP2 and HA-tagged ACK1 or kdACK1 expressing constructs,

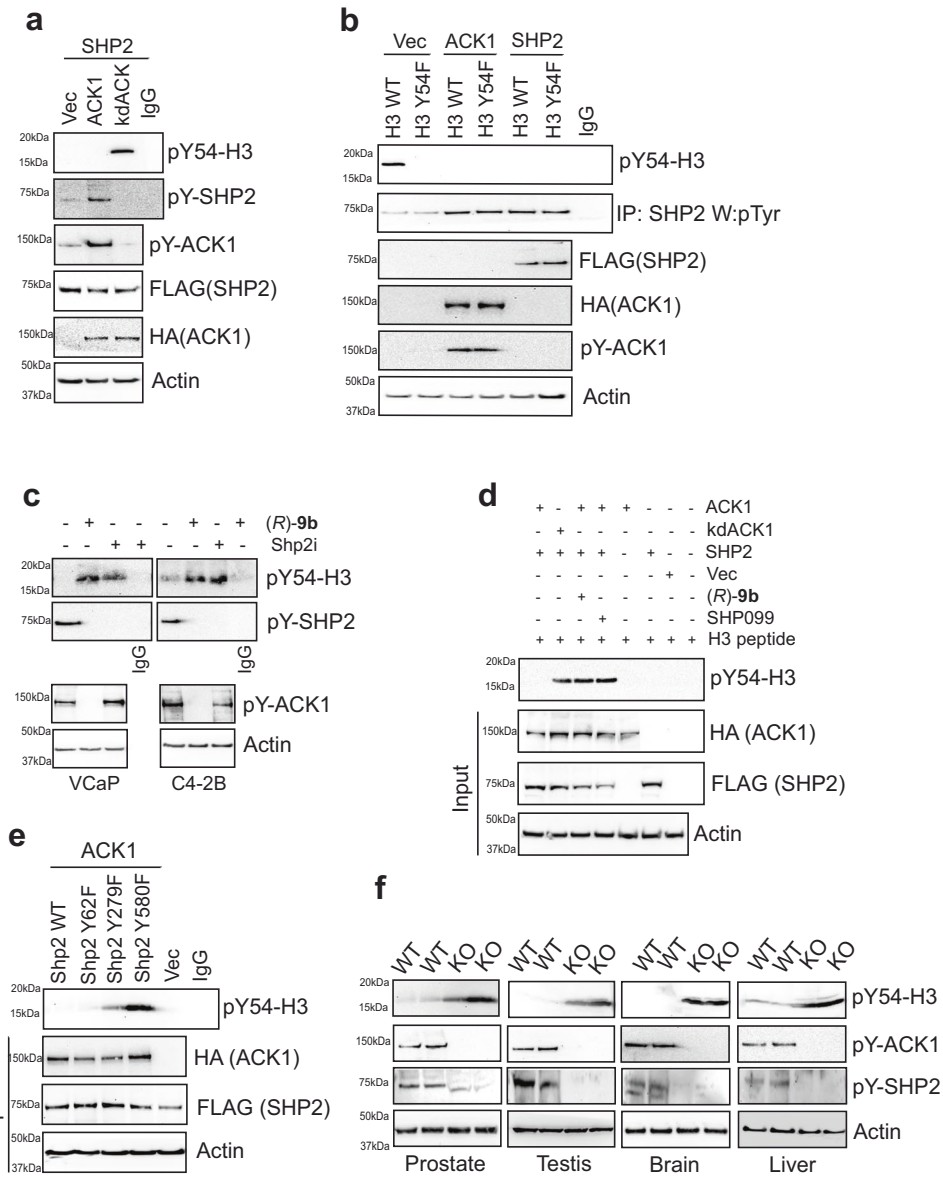

**Fig. 1 | SHP2 erase pY54-H3 epigenetic marks upon its activation by ACK1.**
**a** HEK293T cells were co-transfected with Vector (Vec), FLAG-tagged SHP2 and HA-tagged ACK1 or kdACK1 expressing constructs. Lysates were immunoprecipitated (IP) with pY54-H3, followed by immunoblotting with H3 antibodies (top panel). Lower panels are immunoblots with the indicated antibodies. **b** VCaP cells were co-transfected with FLAG-tagged H3 or H3-Y54F mutant and HA-tagged ACK1 or FLAG-tagged SHP2 expressing constructs. Lysates were IP with pY54-H3, followed by immunoblotting with FLAG antibodies (top panel). Lower panels are immunoblots with the indicated antibodies. **c** VCaP and C4-2B cells were treated with 1 μM of (R)-9b or SHP099 for 8 h and protein lysates were IP with pY54-H3 antibodies followed by immunoblotting with H3 antibody (top panel). Lower panels are immunoblots with the indicated antibodies. **d** HEK293T cells were co-transfected with vector, FLAG-tagged SHP2 and HA-tagged ACK1 or kdACK1 expressing constructs. Post

48 h of transfection, cells were treated with DMSO, (R)-9b or SHP099, and lysates were incubated with beads pre-coated with H3 peptide for 2 h. Immunoprecipitates and inputs were then immunoblotted with pY54-H3, HA, FLAG and Actin antibodies, respectively. **e** VCaP cells were co-transfected with Vec, HA-tagged ACK1 and FLAG-tagged SHP2 or mutants (Y62F, Y279F and Y580F). Lysates were IP with pY54-H3 antibodies, followed by immunoblotting with H3 antibody (top panel). Lower panels are immunoblots with the indicated antibodies. **f** Tissue lysates were prepared from prostates, testis, brain and livers of WT and *Ack1/Tnk2* KO mice. Lysates were IP with pY54-H3 antibodies, followed by immunoblotting with H3 antibody (top panel). Lower panels are immunoblots with the indicated antibodies. For **a-f**, representative images are shown from *n* = 3 biologically independent experiments. Source data are provided as a Source Data file.

followed by treatment with (R)-9b or SHP099. Lysates were subjected to pulldown with beads pre-coated with H3 peptide, followed by immunoblotting with pY54-H3 antibodies. Expression of kdACK1 or loss ACK1 kinase or SHP2 activity resulted in increased pY54-H3 expression (Fig. 1d), suggesting ACK1/SHP2 signaling negatively regulates pY54-H3 levels.

To identify the precise signaling event, Tyr-phosphorylated proteins were enriched from ACK1 overexpressing cells, followed by mass spectrometry. This unbiased screen revealed multiple peptides with known ACK1 auto-phosphorylation sites, including Tyr284, a primary phosphorylation site. In addition, three phospho-peptides derived from SHP2 were identified, indicating SHP2 phosphorylation at Tyr62,

Tyr279 and Tyr580 was targeted by ACK1 (Supplementary Fig. S3a–c). SHP2 mutation of either Y542 or Y580 resulted in reduction of phospho-p44/42 MAPK to baseline levels with a similar decrease in the double Y542/Y580 double mutant[39]. Further, Y279 mutation resulted in reduced SHP2 catalytic activity[47]. SHP2 Tyr62-phosphorylation was primarily observed in tumor tissues, wherein the phosphorylation-mimicking Y62D mutant was 2.4 times more active than the wild-type SHP2, however, the Y62F mutant retained most phosphatase activity[48]. To examine the consequence of targeting by ACK1, we generated SHP2 point mutants, Y62F, Y279F and Y580F, and its relevance to modulating pY54-H3 was assessed. Both, Y580F- and Y279F-SHP2 mutants exhibited increase in pY54-H3 levels, in contrast, pY54-H3 levels were undetectable in the Y62F mutant (Fig. 1e, top 2 panels). Together, these data indicates that ACK1 regulates SHP2 mediated H3 Tyr54-dephosphorylation activity.

To explore the significance of the ACK1/SHP2/pY54-H3 signaling nexus in vivo, we generated a viable *Ack1*/*Tnk2* conditional knockout (*Ack1* KO) mouse model[49]. *Ack1*^flx/wt mice were bred with EIIa-Cre mice to obtain homozygous KO mice that exhibit loss of *Ack1* expression in nearly all tissues (*Ack1* KO mice here onwards). Various organs were isolated from *Ack1* KO mice; a significant increase in pY54-H3 was seen in the prostate, testis, liver, and brain of *Ack1* KO mice (Fig. 1f). In addition, flow cytometric analysis of pY54-H3 was performed, which also revealed a significant increase in its levels in *Ack1* KO mice derived organs (Supplementary Fig. S3d–h), suggesting that ACK1 negatively regulates pY54-H3 levels.

## SHP2 epigenetic activity is dependent on androgen receptor

We reasoned that for its chromatin binding, SHP2 may interact with a protein that undergoes nuclear localization. ACK1 interacts with Androgen receptor (AR)[50], a transcriptional coactivator that plays a paramount role in the onset and progression of prostate cancer. Obligate dependence on AR signaling has led to the development of the AR-antagonists, e.g. Enzalutamide (Enz) and Abiraterone (Abr) that prevent AR nuclear entry and inhibit androgen synthesis thus compromising AR activity, respectively[51,52]. Co-immunoprecipitation studies revealed SHP2/AR complex, which was significantly decreased not only upon AR antagonist treatment, but also upon ACK1 and SHP2 inhibitors treatment (Fig. 2a, panel 2). Nuclear/cytosolic fractionation studies confirmed the presence of pSHP2 in the nucleus and treatment with AR-antagonists or ACK1 or SHP2 inhibitors, compromised nuclear pSHP2 levels (Fig. 2b, top panel). Further, mutant SHP2 were co-expressed with ACK1 and AR, followed by fractionation. While pSHP2 accumulated in the nucleus, SHP2 mutant Y62F exhibited a modest decrease, the Y279F mutant exhibited significant decrease, and the Y580F mutant SHP2 almost completely failed to enter the nucleus (Fig. 2c).

To further validate ACK1/SHP2/AR/pY54-H3 signaling, AR expressing prostate cancer cells were transfected with siRNAs; a significant build-up of pY54-H3 levels was seen upon ACK1, SHP2 or AR knockdown (Fig. 2d). Immunofluorescent studies reveal that the nuclear localization of SHP2 is dependent on both, ACK1 kinase activity directed at the Tyr580 site, as well as AR (Supplementary Fig. 4a, b). However, SHP2 Y580F mutant that exhibited compromised nuclear translocation, showed higher pY54-H3 marks deposition (Supplementary Fig. 4c–f). Together, these data indicates that ACK1 promotes pSHP2/AR complex formation and nuclear translocation consequently expunging pY54-H3 epigenetic marks.

## SHP2 negatively regulates deposition of pY54-H3 epigenetic marks in the *AR* gene

To decipher the pY54-H3 epigenetic footprint, chromatin was prepared from prostates of the mice treated with (*R*)-**9b**. In addition, chromatin was also prepared from prostates of the *Ack1* KO and WT mice, followed by immunoprecipitation with pY54-H3 antibodies and next-generation sequencing (ChIP-seq). Venn diagram (VD) analysis

revealed deposition of pY54-H3 at 805 unique sites in KO prostates, with 99 sites shared between WT and KO prostates, indicating that the large majority of pY54-H3 marked sites did not overlap (Supplementary Fig. S5a and Supplementary data 1). The pY54-H3 peaks were primarily annotated in intergenic regions, and introns of protein-coding genes (Supplementary Table 1). One of the top peaks marked with pY54-H3 was in AR exon, in the prostates of (*R*)-**9b** treated mice, while, another pY54-H3 marking was observed in the enhancer of the *AR* gene, in the prostates of the *Ack1* KO mice (Supplementary data 1). To identify equivalent pY54-H3 markings in the human prostate, LNCaP cells were treated with (*R*)-**9b** and SHP099, followed by sequencing (ChIP-seq). In comparison to vehicle treated sample, (*R*)-**9b** and SHP099 treated samples exhibited enhanced pY54-H3 deposition at two distinct locations, at the *AR* exon1 (nt 66765854-66766158) and 2 (nt 66765242-66765546) (Supplementary Fig. S5b, c).

The pY54-H3 motifs predicted by HOMER, including the corresponding relative score, sequence, and transcription factors are shown in Supplementary Fig. S6a–c. It shows a distinct set of pY54-H3 deposition in the motifs recognized by transcription factors, including a Nur77/NR4A1 recognition sequence motif, which was shared among prostates of *Ack1* KO mice and LNCaP cells treated with (*R*)-**9b**. NR4A1 is nuclear transcription factor belong to steroid-thyroid hormone-retinoid receptor superfamily. Similarly, Nur77 is also a nuclear receptor that acts as a central regulator of T cell immunometabolism, controlling oxidative phosphorylation and aerobic glycolysis during T cell activation[53]. Overall, these data indicates that in addition to AR, chromatin binding (and transcriptional outcome) of other steroid hormone receptors and nuclear receptors that regulate T cell immunometabolism could also be regulated by pY54-H3 epigenetic marks. EnrichR analysis of biological processes regulated by pY54-H3 in LNCaP cells and in prostates of *Ack1* KO mice is shown in Supplementary Figs. S7 and 8.

To validate the deposition of the pY54-H3 marks, ChIP was performed, followed by real-time PCR (ChIP-qPCR) using *AR* exon1 and 2 site-specific primers. Deposition of pY54-H3 marks were specifically enhanced at the *AR* exon1 and 2 upon (*R*)-**9b** or SHP099 treatment in AR positive VCaP, C4-2B and LAPC-4 cells (Fig. 3a–c). Deposition of pY54-H3 marks at control (gene desert) site was minimal. To validate pY54-H3 marking of the *AR* locus, cells were transfected with constructs expressing ACK1, or kinase dead mutant kdACK1, followed by ChIP with pY54-H3 antibody and qPCR. Overexpression of kdACK1 significantly increased pY54-H3 marks deposition at the *AR* locus (Fig. 3d and Supplementary Fig. S9a). To explore the role of SHP2 activity in pY54-H3 regulation, VCaP cells were transfected with SHP2 or mutant expressing constructs. Compared to SHP2, the SHP2-Y279 mutant with reduced catalytic activity exhibited enhanced deposition of pY54-H3 marks at *AR* exons 1 and 2, which was further increased in cells expressing the SHP2-Y580F mutant (Fig. 3e and Supplementary Fig. S9b). To further examine specific deposition of pY54-H3 marks at the *AR* locus, cells were transfected with constructs expressing FLAG-tagged histone H3, or mutant Y54F-H3, treated with (*R*)-**9b** or SHP099, followed by ChIP with pY54-H3 antibody and qPCR. The deposition of pY54-H3 marks at the *AR* locus was significantly compromised when Y54F-H3 mutant was overexpressed (Fig. 3f and Supplementary Fig. S9c).

To assess the role of AR, LAPC4 cells were treated with AR antagonists. Both, Enzalutamide and Abiraterone treatments caused a significant increase in deposition of pY54-H3 marks (Fig. 3g), opening the possibility that AR could be involved in optimal execution of SHP2 epigenetic activity. Earlier, others and we had identified multiple AR phosphorylation sites Tyr267, Tyr307, Tyr363 and Tyr534, targeted by kinases e.g. SRC and ACK1[32,50,54]. In contrast to AR, overexpression of AR-Y267F, -Y307F, -Y534F and Y223/363 F mutants revealed a significant increase in pY54-H3 levels (Supplementary Fig. 10a), which was primarily due to the failure of mutant-AR/SHP2 complex to translocate

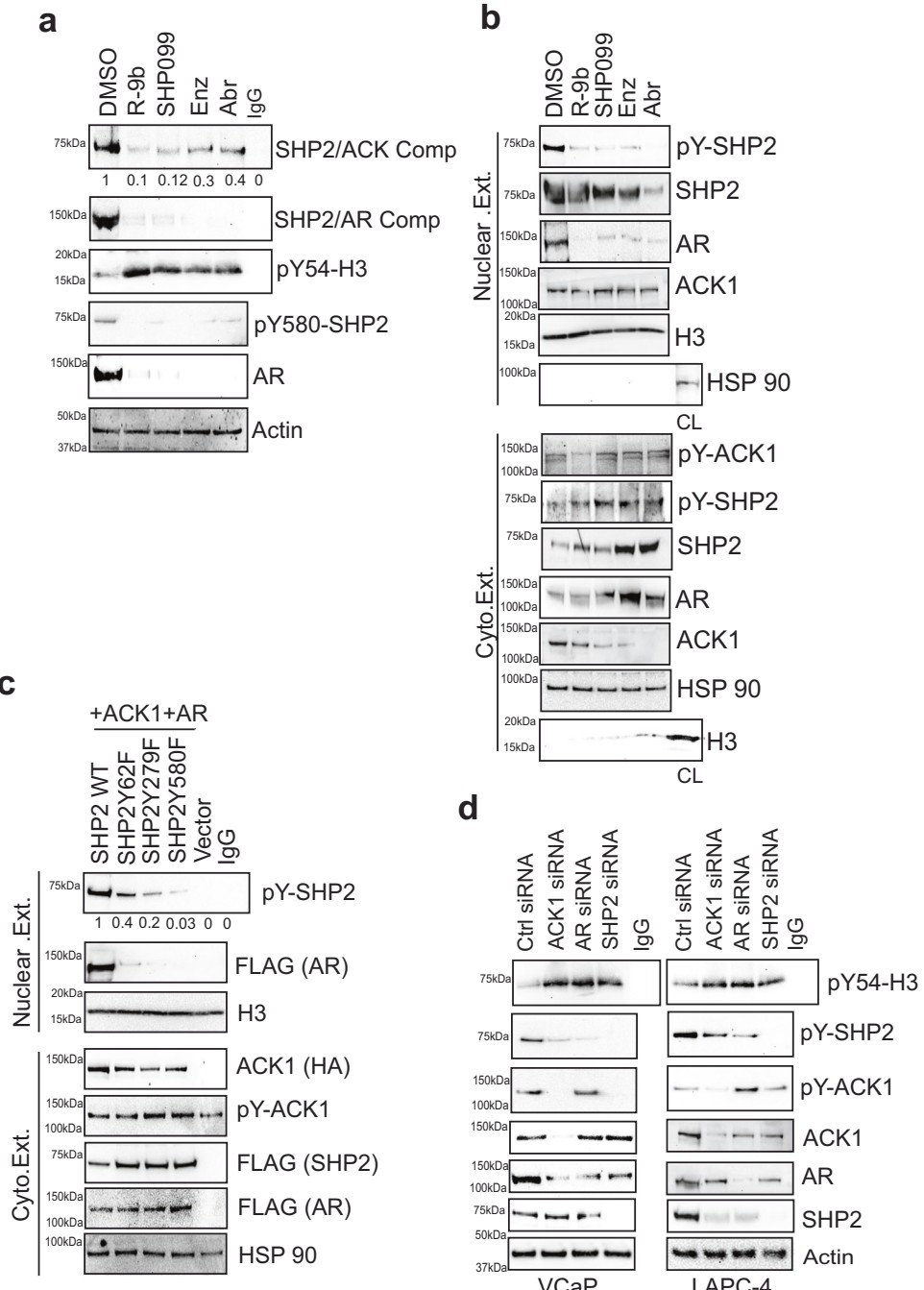

**Fig. 2 | SHP2 epigenetic activity is dependent on AR for its nuclear translocation. a** VCaP cells were treated with 1 μM of (*R*)-**9b**, SHP099, Enz or Abr for 8 h and lysates were IP with SHP2 antibodies, followed by immunoblotting with ACK1 antibody (top panel). Lysates were also IP with pY54-H3 antibodies, followed by immunoblotting with H3 antibody (2nd panel). Lower panels are immunoblots with the indicated antibodies. **b** VCaP cells were treated as above and nuclear and cytosolic extracts were prepared. The lysates were immunoblotted with the indicated antibodies. **c** HEK293T cells were transfected with FLAG-tagged SHP2 or mutants expressing constructs with ACK1 and AR for 48 h and nuclear and cytosolic fractionation was performed. Nuclear lysates were IP with FLAG antibodies, followed by immunoblotting with pTyr antibody (top panel). Lower panels are immunoblots with the indicated antibodies. **d** VCaP, and LAPC-4 cells were transfected with *ACK1/TNK2*, *AR* and *PTPN11* siRNAs and lysates were IP with pY54-H3 antibody, followed by immunoblotting with H3 antibody (top panel). Lower panels are immunoblots with the indicated antibodies. For **a**–**d**, representative images are shown from *n* = 3 biologically independent experiments. Source data are provided as a Source Data file.

into nucleus (Supplementary Fig. 10b). This was further reflected in increased deposition of pY54-H3 epigenetic marks at *AR* exons 1 and 2 when AR was mutated (Fig. 3h and Supplementary Fig. S10c). Moreover, ACK1, SHP2 or AR knockdown by siRNA too caused increased accumulation of pY54-H3 marks at *AR* exons 1 and 2 (Fig. 3i, and Supplementary Fig. S10d, e). Furthermore, ChIP revealed SHP2 binding to AR exon1 and 2 (Supplementary Fig. S10f). Collectively, these data

suggest that pACK1/pSHP2/pAR signaling expunges pY54-H3 epigenetic marks at the *AR* locus.

### ACK1/SHP2/pY54-H3 signaling regulates *AR* and its target gene expression

To examine the transcriptional outcome of the pY54-H3 epigenetic marks deposition, total RNA was prepared from VCaP cells treated

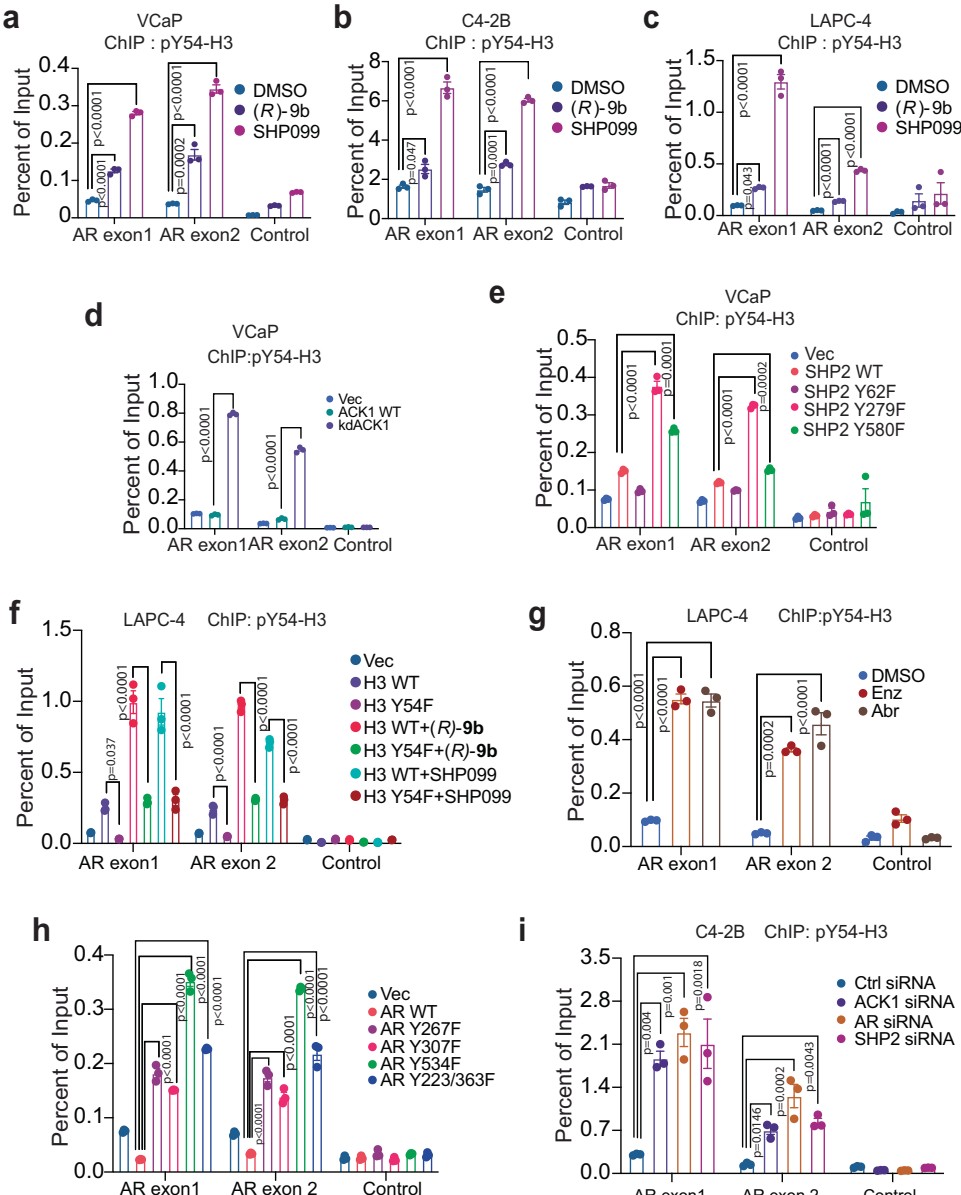

**Fig. 3 | SHP2 erases pY54-H3 epigenetic marks at the *AR* gene locus.** **a**–**c** VCaP, C4-2B and LAPC-4 cells were treated with 1 µM of (*R*)–**9b** or SHP099 for 8 h and lysates were subjected to ChIP with pY54-H3 antibody, followed by qPCR using primers corresponding to *AR* exon 1, 2, or control (IGX, gene desert on chromosome 12) region. **d** VCaP cells were co-transfected with HA-tagged ACK1 or kdACK1 expressing constructs and lysates were subjected to ChIP using pY54-H3 antibody, followed by qPCR using primers corresponding to *AR* exon 1, 2 or control region. **e** VCaP cells were co-transfected with HA-tagged ACK1 and FLAG-tagged SHP2 or SHP2-Y62F, -Y279F or -Y580F mutants. The lysates were subjected to ChIP using pY54-H3 antibody, followed by qPCR using primers corresponding to *AR* exon 1, 2 or control region. **f** LAPC-4 cells were transfected with FLAG-tagged H3 or H3-Y54F mutant and treated with 1 µM of (*R*)−**9b** or SHP099 for 8 h and lysates were subjected to ChIP with pY54-H3 antibodies, followed by qPCR using primers corresponding to *AR* exon 1, 2, or control region. **g** LAPC4 cells were treated with 1 µM of Enz or Abr for 8 h and ChIP was performed using pY54-H3 antibody, followed by qPCR using primers corresponding to *AR* exon1, 2 or control region. **h** LAPC4 cells were co-transfected with HA-tagged ACK1 and FLAG-tagged AR or AR-Y267F, -Y307F, -Y534F, -Y223/363 F expressing constructs. ChIP was performed using pY54-H3 antibody, followed by qPCR using primers corresponding to *AR* exon 1, 2 or control region. **i** C4-2B cells were transfected *TNK2*, *AR* and *PTPN11* siRNA and ChIP was performed using pY54-H3 antibody, followed by qPCR using primers corresponding to *AR* exon 1, 2 or control region. For **a**–**i**, data are represented as mean ± SEM (*n* = 3 biologically independent experiments, three replicates). For **a**–**c**, **e**–**i**, p values were determined by one-way ANOVA. For **d**, p values were determined by unpaired two-tailed Student's *t*-test. p values are shown on the graph. Source data are provided as a Source Data file.

with (*R*)-**9b** and SHP099, followed by qRT-PCR. Both AR and its transcriptional target PSA (*KLK3*) mRNA expression was significantly compromised upon inhibitor treatment (Fig. 4a, b). Transfection with kdACK1 caused a significant decrease in AR and PSA mRNA expression (Fig. 4c, d and Supplementary Fig. S11a). Moreover, cells transfected with SHP2-Y279F and Y580F mutants too caused a significant decrease in AR and PSA mRNA expression (Fig. 4e, f and Supplementary Fig. S11b). Additionally, depletion of ACK1, SHP2 or

AR by siRNA also caused decreased AR and PSA mRNA expression (Fig. 4g, h).

To determine a direct role for pY54-H3 in repressing in AR expression, VCaP and LAPC4 cells were transfected with H3- and Y54F-H3-expressing constructs. An increase in AR and PSA mRNA expression was seen in Y54F-H3 expressing constructs (Fig. 4i, j and Supplementary Fig. S11c). Together, these data indicate that pY54-H3 is a repressive epigenetic mark, and ACK1/SHP2 signaling mediated its

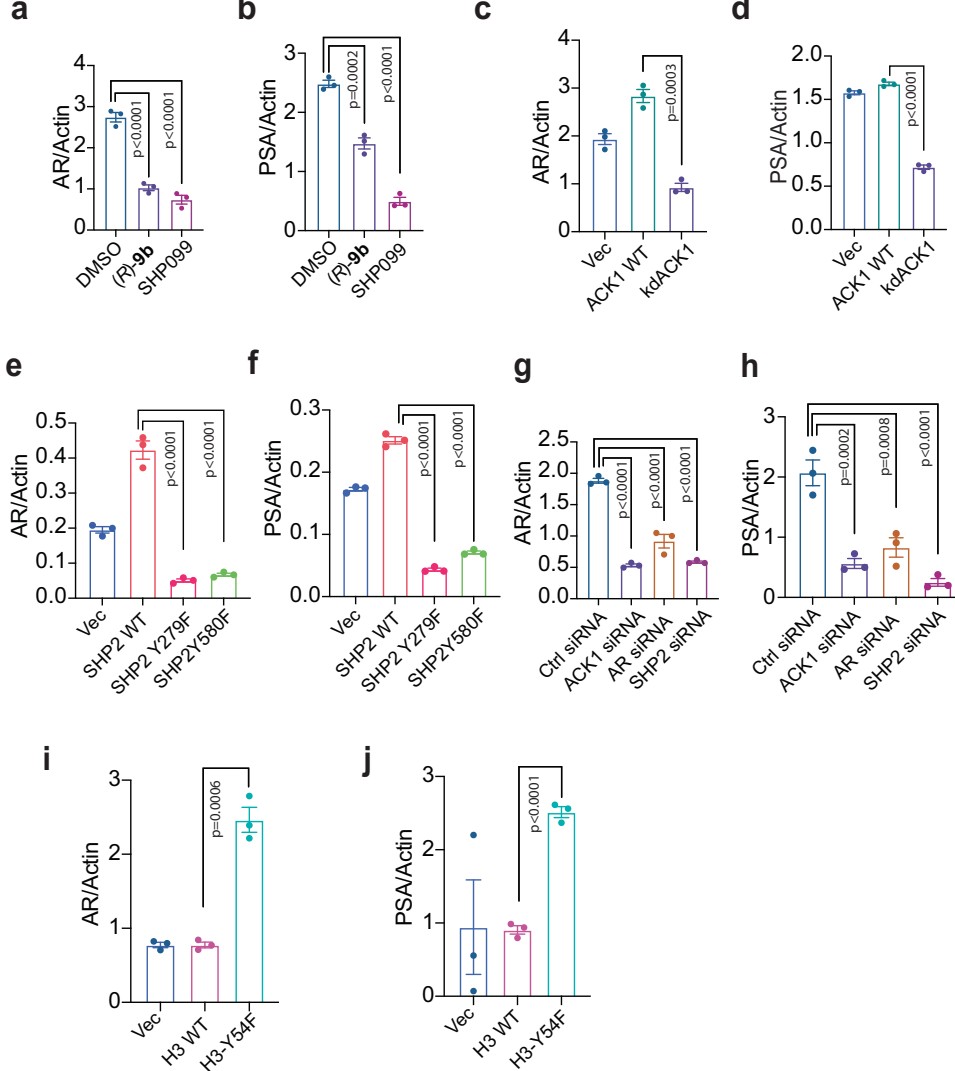

**Fig. 4 | Enhanced pY54-H3 epigenetic marks suppresses *AR* and *PSA* transcription. a, b** VCaP cells were treated with vehicle (DMSO), (*R*)-**9b** or SHP099 for 18 h. Total RNA was isolated followed by qRT-PCR with AR, PSA/KLK3 or actin primers. **c, d** Total RNA isolated from VCaP cells transfected with ACK1 or kdACK1 constructs for 48 h, and subjected to qRT-PCR with AR, PSA or actin primers. **e, f** Total RNA isolated from VCaP cells transfected with SHP2 or SHP2 mutant constructs for 48 h, and subjected to qRT-PCR with AR, PSA or actin primers. **g, h** Total RNA isolated from VCaP cells transfected with *TNK2*, *AR* and *PTPN11* siRNAs for 48 h, and subjected to qRT-PCR with AR, PSA or actin primers. **i, j** Total RNA isolated from VCaP cells transfected with H3 or Y54-H3 mutant expressing constructs, and subjected to qRT-PCR with AR, PSA or actin primers. For **a–j**, data are represented as mean ± SEM (*n* = 3 biologically independent experiments, three replicates). For **a, b, e–h**, p values were determined by one-way ANOVA. For **c, d, i, j**, p values were determined by unpaired two-tailed Student's *t*-test. p values are shown on the graph. Source data are provided as a Source Data file.

deposition and suppress the transcriptional outcome of AR and its target genes.

## NCORs act as readers of pY54-H3 epigenetic marks, suppressing transcription

AR activity is determined by multiple coactivators and corepressors, which it assembles on the promoters of *AR* regulated genes. Various acetyltransferases such as CBP, p300, PCAF, TIP60, and ARD1 acetylate AR, and functionally regulate the transcriptional activity of the AR[33,55], while NCOR1 and NCOR2/SMRT are AR corepressors[56,57]. We reasoned that being a repressive epigenetic mark, pY54-H3 could recruit a nuclear receptor corepressor to downregulate the transcription. Protein lysate was prepared from (*R*)-**9b** treated LNCaP cells, followed by affinity pull down of proteins bound to biotinylated pY54-H3 peptide immobilized on streptavidin beads. The beads were washed, and the bound proteins were identified by mass spec analysis. It led to identification of NCOR2 protein that tightly bound to pY54-H3 epigenetic

marks (Supplementary data 2). To validate NCOR as an epigenetic reader of pY54-H3 marks, biotinylated Y54-phospho-peptide derived from histone H3 (49-63 aa) was generated. As a control, biotinylated Y176-phospho-peptide derived from AKT (167-181 aa) was used. Pull down with streptavidin beads followed by immunoblotting indicated that both NCOR1 and NCOR2 specifically bound to pY54-H3 peptide (Fig. 5a). Further, ChIP was performed using chromatin isolated from VCaP and C4-2B cells that were treated with (*R*)-**9b** or SHP099, confirming NCOR1 and NCOR2 binding to *AR* exons 1 and 2, which were marked with pY54-H3 marks (Fig. 5b, c, and Supplementary Fig. S12a).

The N-terminus of NCOR1 and NCOR2 has been characterized to contain three independent repression domains (RD1, RD2 and RD3) and SW13/ADA2/NCOR/TFIIB (SANT) domain, two copies of which are located between RD1 and RD2 (Fig. 5d). The first SANT domain is referred to as the deacetylase activation domain (DAD), which directly binds to HDAC3, and the second SANT domain acts as a histone interaction domain (HID)[58]. We generated a construct expressing His-

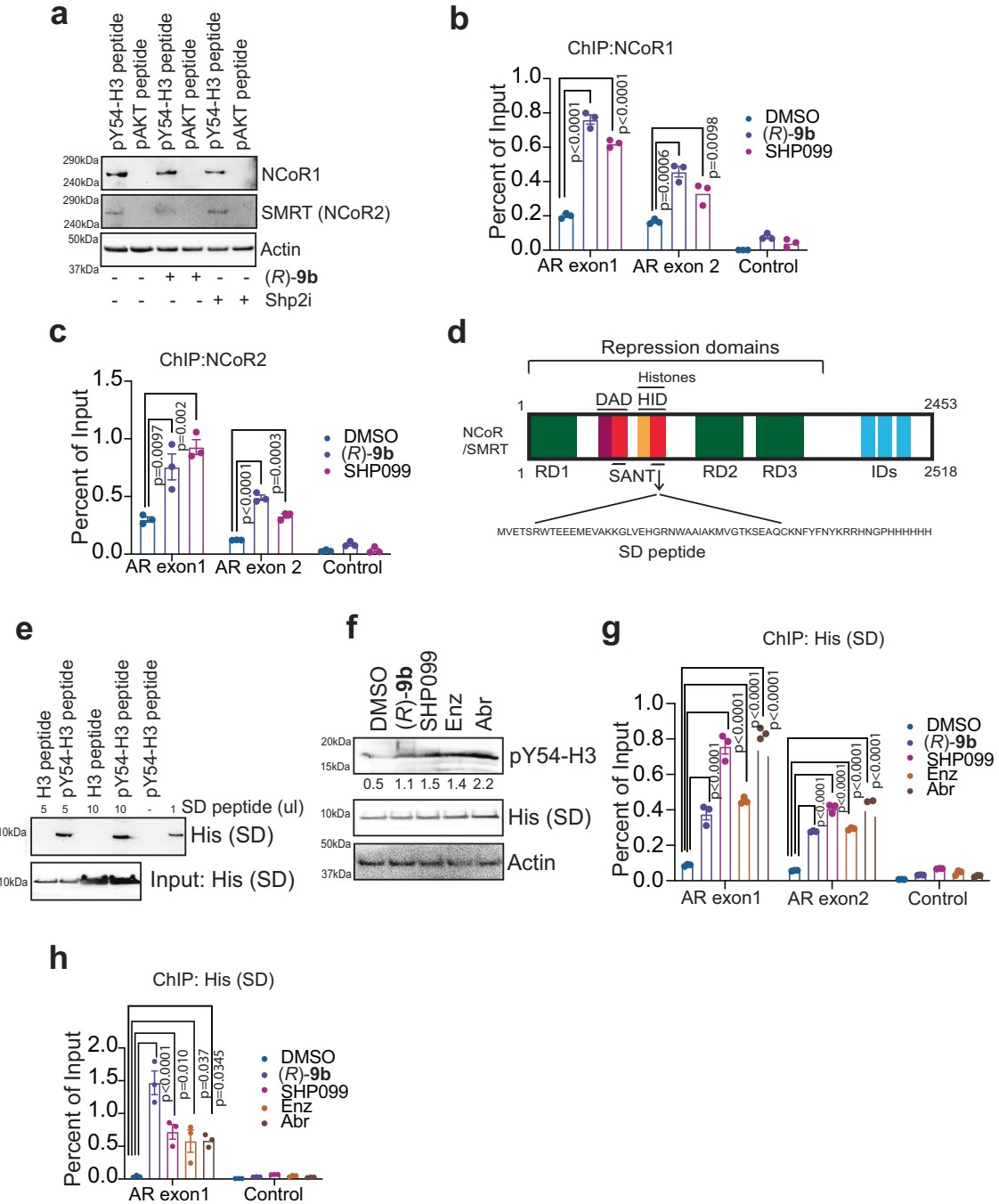

**Fig. 5 | NCOR1 and 2 are epigenetic readers of pY54-H3 marks. a** VCaP cells were treated with 1 μM of (R)-**9b**, SHP099 for 18 h and lysates were IP with pY54-H3 or pAKT peptides bound to streptavidin beads, followed by immunoblotting with NCOR1, NCOR2 (SMRT) or actin antibodies, respectively. **b, c** C4-2B cells were treated with 1 μM of (R)-**9b**, SHP099 for 16 h and ChIP was performed using NCOR1or NCOR2 (SMRT) antibodies, followed by qPCR using primers corresponding to *AR* exon 1, 2 or control region. **d** Pictorial representation of NCOR/SMRT corepressors showing deacetylase activating domain (DAD), histone interaction domain (HID), SW13/ADA2/NCOR/TFIIB (SANT), repression domains (RD1, 2 and 3), and nuclear receptor interaction domains (IDs) involved in interactions with nuclear receptors. **e** In vitro binding assay was performed with biotinylated pY54-H3 or H3 peptides that were incubated and with varying concentration of His-tagged SANT domain (SD) peptide. Pull-down was performed with streptavidin beads, followed by immunoblotting for detection of His-tagged peptide. **f** VCaP cells were treated with 1 μM of (R)-**9b**, SHP099, Enz, or Abr for 18 h and lysates were prepared. These lysates were then immunoprecipitated with His-tagged SD peptide bound to Ni-NTA resin, and then immunoblotted with pY54-H3 antibodies. **g, h** VCaP and C4-2B cells were transfected with His-tagged SD expressing constructs for 48 h and treated with 1 μM of (R)-**9b**, SHP099, Enz, or Abr for 18 h. Lysates were prepared, and ChIP was performed using Ni-NTA resin, followed by qPCR using primers corresponding to *AR* exon 1, 2 or control region. For **a, e**, and **h**, representative images are shown from n = 3 biologically independent experiments. For **b, c, g**, and **h** data are represented as mean ± SEM (n = 3 biologically independent experiments, three replicates). p values were determined by one-way ANOVA. p values are shown on the graph. Source data are provided as a Source Data file.

tagged SANT domain (52 aa long, SD peptide) (Fig. 5d). In vitro binding assay revealed specific binding of SD peptide with the phosphorylated-Y54-H3 peptide, while sparing unphosphorylated H3 peptide (Fig. 5e). Further, pull-down was performed using His-beads bound to SD peptide, revealing a significant increase in SANT/pY54-H3 complex formation upon (R)-**9b**, SHP099, Enzalutamide and Abiraterone treatment (Fig. 5f). Moreover, ChIP confirmed the SANT domain of NCOR1/2 binding to *AR* exon 1 and 2, which were marked with pY54-H3 marks by the (R)-**9b**, SHP099, Enzalutamide and abiraterone treatments (Fig. 5g, h). Furthermore, pY54-H3 marks deposition are not dependent on NCOR1 and NCOR2 as siRNA knockdown did not affect the pY54-H3 levels (Supplementary Fig. S12b). Taken together, these results indicate that the SANT domain of the NCORs is an epigenetic reader of pY54-H3 marks that promotes transcriptional repression.

### pY54-H3 depletion correlates with prostate cancer progression and targeting the pACK1/pSHP2 axis mitigates tumors by restoring pY54-H3

Both activated ACK1 (pY284-ACK1 or pACK1) and AR levels increases as prostate cancer progresses from the benign prostatic hyperplasia (BPH) to the late stages of prostate cancer including the metastatic stage[28,59]. To assess the correlation between the levels of pY580-SHP2 and pY54-H3 in the successive stages of the prostate cancer, a tissue microarray (TMA) of clinically annotated human prostate (n = 80) samples was analyzed. TMAs were stained with pY580-SHP2 and pY54-H3 antibodies. Increased pY580-SHP2 expression was observed when cancer patients from progressive stages (stage I to IV) were examined (Fig. 6a, b), with significantly higher expression in stage IV, as compared to stages I to III (Supplementary Fig. S12c). The expression of pY580-SHP2 was also significantly correlated with prostate cancer with increasing Gleason scores, from 2 to 5 (Supplementary Fig. S12d). In contrast, a significant decrease in expression of pY54-H3 marks was seen as prostate cancer progressed to malignant stage IV (Fig. 6a, c). Further, expression of pSHP2 and pACK1 was inversely correlated with pY54-H3 in situ (Spearman rank correlation coefficient R = −0.59, p = 2.1E-08; R = −0.25, p = 0.025, respectively, Fig. 6d, e). Furthermore, expression of pACK1 was directly correlated with the pSHP2 (Fig. 6f), indicating that expression of pACK1/pSHP2 positively and pY54-H3 negatively correlates with progression of prostate cancer to metastatic stage.

To investigate the clinical relevance of pACK1/pSHP2/pY54-H3/AR signaling, VCaP cells were implanted subcutaneously in male SCID mice. When the tumors reached approximately 100 mm³ in size, the mice were randomized and given oral gavage with either the vehicle, ACK1 inhibitor (R)-**9b**, SHP2 inhibitor, SHP099, or Enzalutamide. Although vehicle-treated mice formed robust CRPC tumors, tumor growth was compromised in (R)-**9b**, SHP099, and Enzalutamide treated mice, with the smallest tumors in ACK1 and SHP2 inhibitor-treated mice (Fig. 7a–c). Further, tumors were excised, RNA was prepared, and immunoblotting was performed. A significant increase in pY54-H3 marks was observed in the tumors from (R)-**9b**, SHP099, and Enzalutamide-treated mice, while the AR protein expression was significantly downregulated (Fig. 7d, top 2 panes). Increase in pY54-H3 marks in the tumors from (R)-**9b**, SHP099, and Enzalutamide-treated mice was also reflected in the significant downregulation of *AR* and *PSA* expression (Fig. 7e, f). Thus, pACK1 activity is critical for SHP2 phosphatase activity that erase pY54-H3 marks, leading to enhanced expression of *AR* and its target genes. Taken together, these data indicates that pACK1/pSHP2 signaling promotes prostate tumors growth by overcoming pY54-H3 marks, causing increased *AR* expression.

### Increased pY54-H3 deposition causes decreased AR expression in *Ack1* knockout, *Shp2* knock-in mice and a NSML patient

NSML is characterized by multiple deformities, including retardation of growth and abnormalities of genitalia, including undescended testes (cryptorchidism) and a urethra that opens on the underside of the penis (hypospadias)[22,26]. Male sexual differentiation, testicular descent, and spermatogenesis requires AR activity. The external genitalia of male *Ar* KO mice show ambiguous or feminized appearance, the urethra shows hypospadias and the testes are small and cryptorchid in the low abdominal area[60]. Phenotypic similarities among the human disorder and mutant mice led us to explore whether these abnormalities are caused by catalytically inactivating mutations in *PTPN11* resulting in epigenetic repression of *AR*. A knock-in mouse harboring the *Ptpn11* mutation Y279C (Y279C/+ or NSML/+), recapitulated the human disorder, with short stature, craniofacial dysmorphia, and hypertrophic cardiomyopathy[61]. Tissue lysates from the prostate, testis and liver of NSML/+ mice exhibited a significant increase in pY54-H3 epigenetic marks as compared to WT mice (Fig. 8a).

ChIP-seq of the prostates of *Ack1* KO (and WT) mice had revealed two pY54-H3 peaks in *Ar*, exon 1 and in the enhancer (Fig. 8b). ChIP followed by real-time PCR revealed enhanced pY54-H3 marks deposition at the *Ar* loci (AR1 and AR2) in the prostate, testis and liver of NSML/+ (LS) mice (Fig. 8c). It was also reflected in a significant decrease in the transcription of *Ar* and *Tmprss2*, another *Ar* target gene (Fig. 8d, e). Consistent with these data, immunohistochemcial (IHC) staining of prostates from *Ack1* KO mice and the WT mice treated with (R)-**9b** and SHP099 exhibited significant decrease in pY-Shp2 and increase in pY54-H3 marks, compared to WT prostates (Supplementary Fig. 13a). Further, increased deposition of pY54-H3 at *Ar* loci (Supplementary Fig. 13b), and significant loss of AR and TMPRSS2 mRNA expression was seen in the prostates obtained from *Ack1* KO and the WT mice treated with (R)-**9b** and SHP099 inhibitors (Supplementary Fig. 13c).

An expression of ACK1 is present in hematopoietic cells[49]. To further examine the role of pSHP2/pY54-H3 signaling in NSML, induced pluripotent stem cells (iPSCs) were derived from a NSML patient (Q510E iPSC), and a healthy individual (WT iPSCs). Q510E iPSCs exhibited a robust increase in global pY54-H3 levels (Fig. 8f), as well as their increased deposition at the *AR* gene locus (Fig. 8g), resulting in a significant decrease in *AR* mRNA and protein expression (Supplementary Fig. 14a and Fig. 8f, 2nd panel). In addition, VCaP cells were retrovirally infected with SHP2 NS mutant (D61G), revealing a significant decrease in pY54-H3 levels (Supplementary Fig. 13d).

To examine whether functional rescue of AR activity can be attained, WT and Q510E iPSCs were treated with the AR ligand, Dihydrotestosterone (DHT), followed by quantitation of AR target genes, *FKBP5*, and *ANAPC10*, which are expressed in the hematopoietic milieu[62]. As *AR* expression is self-regulated[28], AR levels were also examined by qRT-PCR. At a low concentration of DHT for 24 h, *AR*, *FKBP5*, and *ANAPC10* mRNA expression was significantly lower in Q510E iPSCs. However, higher (10x) and extended DHT treatment (48 h) caused significant increase in *AR*, *FKBP5*, and *ANAPC10* mRNA expression in Q510E iPSCs, at the levels comparable to expression in WT iPSCs (Supplementary Fig. 14a–c). These results taken together indicate that expression of AR target genes including AR itself may be rescued upon treatment with DHT, indicating a potential therapeutic approach to NSML disease.

## Discussion
Tyrosine phosphorylation is a fundamental mechanism regulating numerous cellular signaling transduction pathways and has been implicated in the etiology of diverse human diseases[1,2,63]. A few kinases have recently been shown to carry out a distinct epigenetic act, Tyr-phosphorylation of histones[28,41,42,64–67]. However, a direct chromatin-modifying role for protein tyrosine phosphatases (PTPs) was not known. We uncovered that SHP2 is a distinct class of an epigenetic enzyme that specifically erases the pY54-H3 repressive marks (Fig. 8h). ACK1 kinase has ability to phosphorylate AR and SHP2, promoting

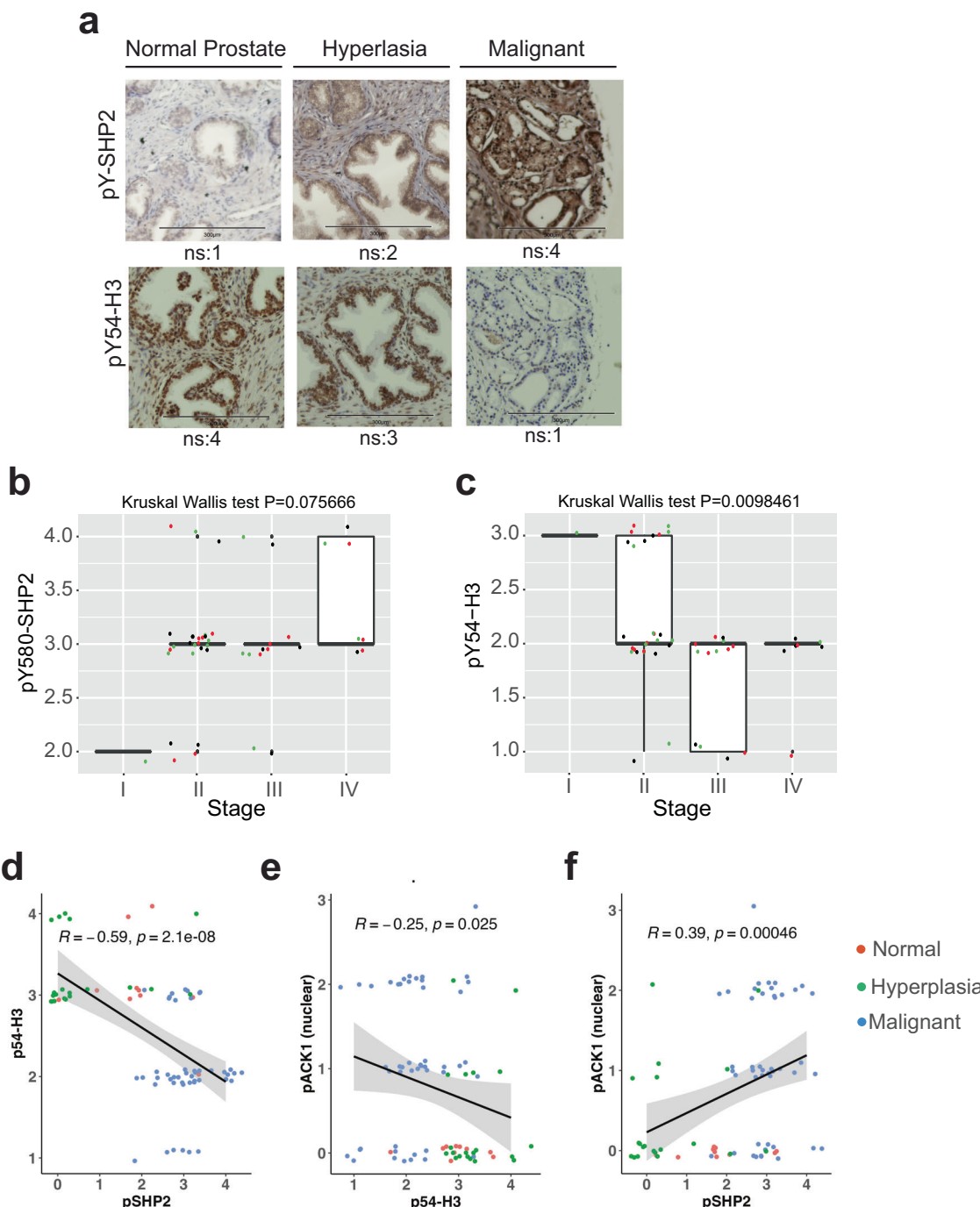

**Fig. 6 | Depletion of pY54-H3 marks and increased pSHP2 levels during prostate cancer progression to malignant stage. a**, **b** Human prostates TMA were subjected to immunohistochemistry with pY580-SHP2 (**a**) and pY54-H3 (**b**) antibodies (*n* = 80 cores per slide). Pathologist's scores for nuclear staining (ns) in each sample is shown. Bars represent 200 μm. **b**, **c** Box plots summarize distributions of staining intensities for pY580-SHP2 and pY54-H3 antibodies in prostate TMA sections. The box indicates 50% of the data from the 25% quartile to the 75% quartile with the bold black horizontal lines representing the median. The individual data points were jittered and colored for better visualization. **d**–**f** Expression levels between pSHP2 and pY54-H3, pACK1 and pY54-H3, and pACK1 and pSHP2 were significantly correlated in prostate tumors. For **b**, **c**, p values were determined by two-sided Kruskal–Wallis test. For **d**, **e**, the black line is the fitted linear line with 95% confidence interval bands shaded in gray. The spearman correlation coefficient was calculated with p values testing the estimated correlation against a null value of zero based on the two-sided correlation test. All p values were not adjusted for multiple comparisons. p values are shown on the graph. Source data are provided as a Source Data file.

SHP2/AR complex formation, which translocates to nucleus, followed by dephosphorylation of pY54-H3 at *AR* locus. Consequent AR transcriptional activation establishes an ACK1/SHP2/pY54-H3/AR signaling nexus that seems to regulate a broad spectrum of fundamental physiological processes linked to human diseases. Mutations of ACK1 and

SHP2 in tumor cells (cBioPortal) shows evidence for ACK1 and SHP2 co-occurrence in multiple malignancies (Supplementary data 3). Remarkably, both ends of the spectrum, loss, and gain of ACK1/SHP2/pY54-H3/AR signaling, are not only associated with distinct disorders but also provide a long-sought explanation of how modulating the

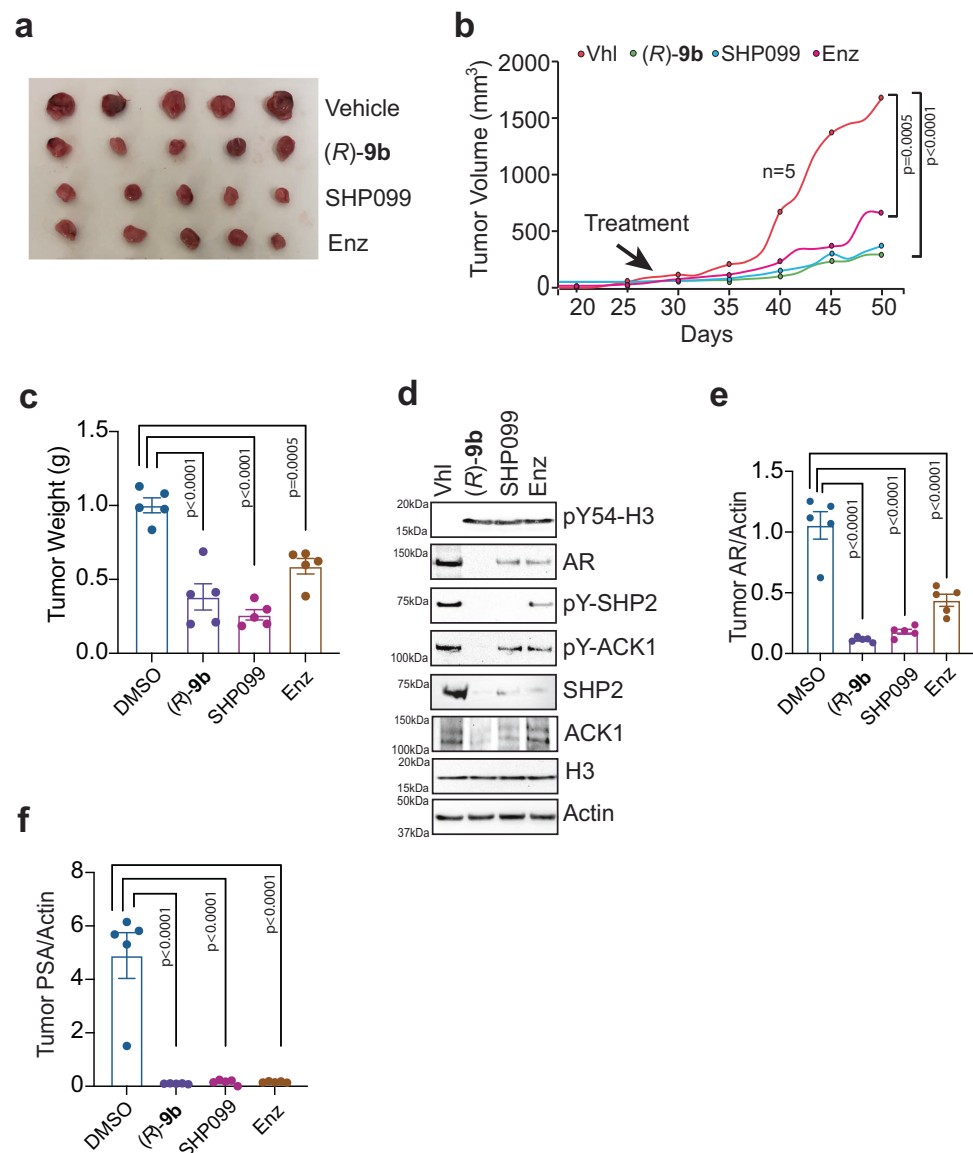

**Fig. 7 | Inhibition of ACK1, SHP2 or AR activity suppresses *AR* transcription and prostate tumor growth by elevating pY54-H3 marks. a** VCaP cells ($2 \times 10^6$/mice) were injected subcutaneously in SCID mice and once tumors were palpable (~100 mm³, in about four weeks), mice were injected with vehicle (6% Captisol), (*R*)-**9b**, SHP099, and Enzalutamide, five times a week. Mice were euthanized, tumors excised, photographed. **b** Tumor volumes were measured using calipers and plotted. **c** Tumor weights were plotted. **d** Tumors lysates were processed for IP with pY54-H3 antibody, followed by immunoblotting with H3 antibodies (top panel), pY-SHP2, pY-ACK1, AR, H3 and actin antibodies, respectively. Representative images are shown from *n* = 3 biologically independent experiments. **e, f** Total RNA was isolated from tumors and subjected to qRT-PCR with AR, PSA and actin primers. For **a–c**, **e**, **f**, *n* = 5 mice in each group. For **b**, **c**, **e** and **f**, data are represented as mean ± SEM. *p* values were determined by one-way ANOVA. p values are shown on the graph. Source data are provided as a Source Data file.

activity of a single phosphatase could indeed be the underlying cause of the diseases. NSML patients, as well as SHP2 knock-in mice, both with loss-of-function mutations, exhibited robust upregulation of pY54-H3 epigenetic marks and a significant decrease in AR expression. However, androgen treatment was able to rescue the compromised AR functionality in iPSCs derived from NSML patients, suggesting that SHP2/pY54-H3 signaling is not only essential for the maintenance of AR homeostasis in normalcy, but also functional AR reactivation could partly compensate for the loss of SHP2 activity loss and potentially overcome phenotypic changes, i.e. genitalia abnormalities and retardation of growth associated with NSML.

AR signaling plays an indispensable role in both the initiation and progression of prostate cancer to the metastatic stage[27,28,30]. Interestingly, AR expression in prostate cancer is positively regulated by another histone phosphorylation mark, pY88-H4[28]. Overall, AR

homeostasis is critically dependent on the balance between two distinct epigenetic marks, the repressive pY54-H3 marks, which is read by NCORs and maintains AR levels in check, and activating pY88-H4 marks, which are read by WDR5/MLL2 complex to upregulate AR expression. Thus, the 'epigenetic switch' from pY54-H3 to pY88-H4 at the AR locus determines expression of the AR global transcriptome, driving prostate tumor growth. Consistent with these data, treatment with ACK1, SHP2 and AR antagonists resulted in a significant increase in pY54-H3 and downregulation of AR expression, compromising prostate tumor growth. Combined with increased phospho-SHP2 and progressive loss of pY54-H3 in patients, these results establish a crucial role for ACK1/SHP2/pY54-H3/AR signaling in prostate cancer pathogenesis. To understand how SHP2 is activated in the nucleus we hypothesized that the two SH2 domains of SHP2 (N- and C terminal) could make contact with Tyr-phosphorylated residues in ACK1, leading to SHP2

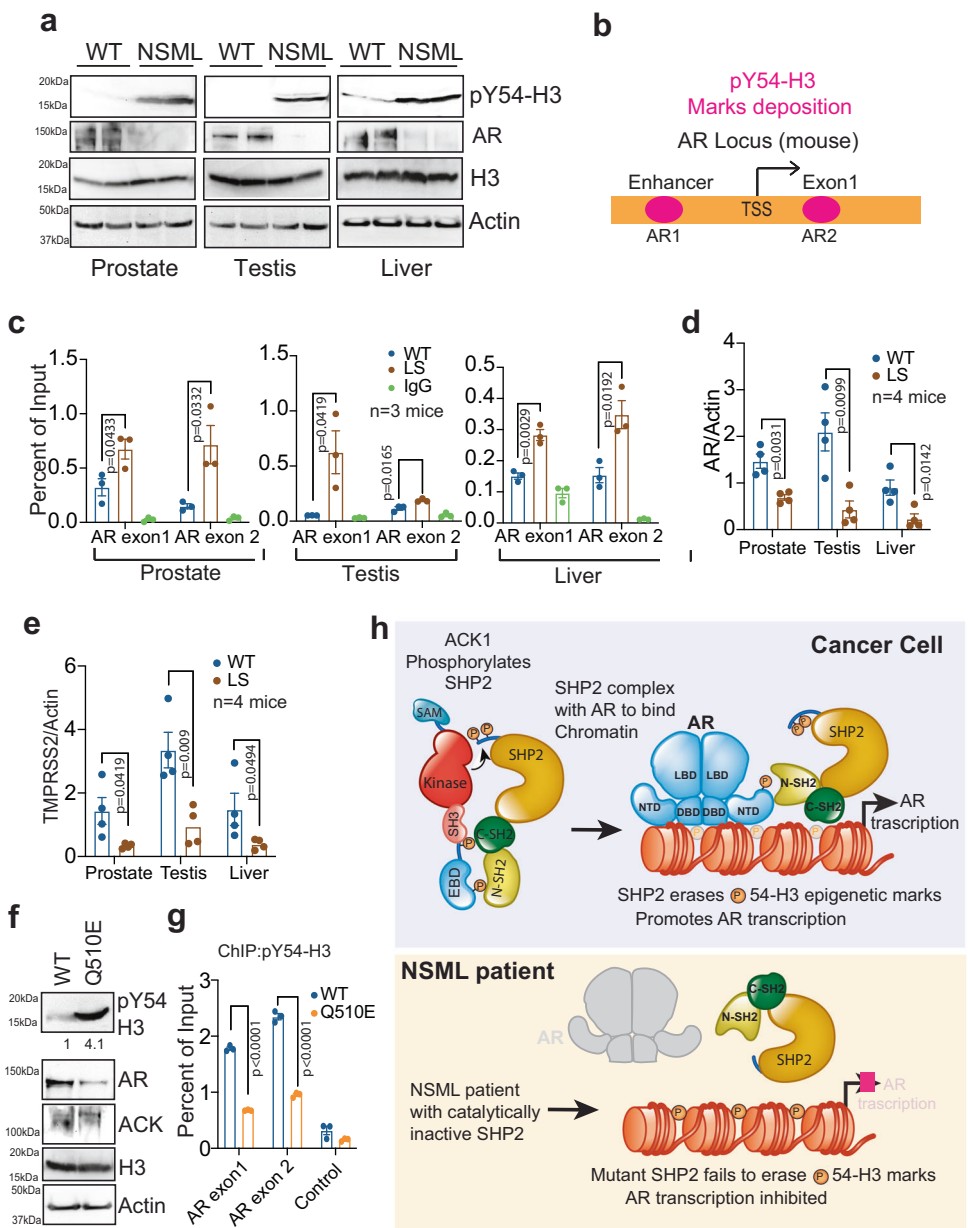

**Fig. 8 | Increased pY54-H3 marks is associated with NSML. a** Lysates from prostate, testis, and liver of NSML/+ mice were immunoprecipitated with pY54-H3 antibodies, followed by immunoblotting with H3 antibodies (top panel). Lysates were also subjected to immunoblotting with actin antibodies (lower panel). **b** ChIP-seq was performed using pY54-H3 antibody and the peaks at enhancer (AR1) and exon 1 (AR2) sites of the mouse *Ar* are shown in graphical format. The numbers indicate nucleotide position at the midpoint of the peaks. **c** Lysates from prostate, testis and liver of NSML/+ (LS) mice were subjected to ChIP using pY54-H3 (or IgG as control) antibody, followed by qPCR using primers corresponding to AR regions 1 and 2. (*n* = 3 mice in each group, 3 replicates). **d, e** Total RNA isolated from prostate, testis and livers of NSML/+ (LS) mice and were subjected to qRT-PCR with AR, TMPRSS2 and actin primers. (*n* = 4 mice in each group, 3 replicates). **f** iPSCs derived from a NSML patient (Q510E iPSC) and the healthy individual (WT iPSCs)

were subjected to IP with pY54-H3 antibody, followed by immunoblotting with H3 antibodies (top panel). Lower panels are immunoblots with the indicated antibodies. **g** iPSCs (derived from a NSML patient) lysates were subjected to ChIP using pY54-H3 antibody (or IgG), followed by qPCR using primers corresponding to *AR* exon 1 and 2. For **a** and **f**, representative images are shown from 3 biologically independent experiments. For **c, d, e, g**, data are represented as mean ± SEM. *n* = 3 biologically independent samples (three replicates). p values were determined by unpaired two-tailed Student's *t*-test. p values are shown on the graph. Source data are provided as a Source Data file. **h** Graphical Abstract: SHP2 phosphorylation by ACK1 promotes interaction with AR, followed by nuclear translocation. SHP2 erases pY54-H3 epigenetic marks upregulating AR transcription, a schematic representation.

open conformation and its enhanced catalytic/phosphatase activity. As per Diop et al. the SH2 domains of SHP2 makes contact with Tyr-phosphorylated residues based on pY-Ψ-x-Ψ motif (Ψ being hydrophobic residue)[68]. We checked all the known Tyr-phosphorylation events in ACK1 and observed that potentially ACK1 pTyr518 (pYDPV) and ACK1 pTyr859/860 (YpYLLP) could make contacts with two SH2

domains of SHP2. Both the SH2 domains in SHP2 (PDB 2SHP) appear to be accessible for phosphoTyr-peptides; ACK1 pY518 and pY859/Y860 fit well with two SH2 domain of SHP2 (Supplementary Fig. 14d–g). Significantly, the sequence between Y518 and Y859/Y860 is Proline-rich, which provides enough flexibility, thus potentially allowing ACK1 pY518 and pY859/Y860 to bind to both SH2 domains simultaneously,

bringing ACK1 kinase domain in vicinity of SHP2 phosphatase domain, facilitating its activation (see the model Fig. 8h).

Consistent with prostate tumors, Ptpn11E76K mutant mice with hyperactivation of SHP2 (which develop NS), exhibited reduction in pY54-H3 levels (our unpublished data). NS mice have an enhanced proinflammatory phenotype, modified resident macrophage home-ostasis, and triggered monocyte infiltration, resulting in inflammation-driven insulin resistance[69]. Overall, these data indicates that pY54-H3 critically regulates functional outcomes of both, the gain and loss of SHP2 phosphatase activity; aberrantly high pY54-H3 may cause imbalance in growth characteristics and development in the male reproductive system, while in contrast, subduing pY54-H3 it could also have implications in metabolic disorders, antitumor immunity, or cancer.

SHP2 activation is known to influence the tumor microenvironment; an activating mutation in *PTPN11* in mesenchymal stem cells exhibited excessive production of CCL3, which recruited monocytes leading to aggravated childhood myeloproliferative neoplasm[70]. In lung cancer, SHP2 inhibition depleted alveolar and M2-like macrophages, and induced tumor-intrinsic CCL5/CXCL10 secretion causing B and T lymphocyte infiltration in tumors[19]. Interestingly, *Ack1* KO too exhibited activation of CD8[+] and CD4[+] T cells, compromising ICB-resistant prostate tumor growth[49]. Further, a significant increase in CXCL10 was observed upon ACK1 inhibitor treatment. Mechanistically, the T cell activation is negatively regulated by C-terminal Src kinase (CSK); ACK1 dampens T-cell response by augmenting CSK activity through its Tyr18-phosphorylation. Thus, although the mechanism promoting T lymphocyte infiltration into tumors upon treatment with ACK1 and SHP2 inhibitor appears to be different, neither studies have explored the role of ACK1/SHP2/pY54-H3/AR signaling axis in immune cells. Considering the ubiquitous expression of ACK1 and SHP2 (and AR in certain compartments) in the immune cells, and ACK1 regulating SHP2 activation, it may prove insightful to examine the role of this signaling nexus in immune cells within the tumor microenvironment.

Can other members of the PTP superfamily of enzymes regulate pY54-H3 levels? Although experimental evidence is currently lacking, inhibitors of PTPN2 (SF1670), PTPN1 (KY-226) and PTPN6 (Sodium stibogluconate) did exhibit significant increase in pY54-H3 levels (Supplementary Fig. 15a). PTPN2 (TCPTP)[71], PTPN6 (SHP1)[72] and PTPN22 (PEP)[73] have presence in the nuclei, while PTPN23 is shown to be present in nuclei of brain[74]. We performed siRNA mediated knockdown and observed increase in pY54-H3 levels, with PTPN11 having the most activity, followed by PTPN6 and PTPN1 (Supplementary Fig. 15b, c). Together, these data indicates that the spatial and temporal regulation of pY54-H3 deposition may be discrete resulting in a distinct transcriptional outcome for these nuclear phosphatases. Follow-up studies could identify the distinct functional relevance of nuclear phosphatases in expunging pY54-H3 marks.

Relevance of pY54-H3 marks is not confined to AR; indeed, we have observed that many other important genes too are also regulated by these marks, which includes *Akt1*, *Akt2*, *Ly6e* and *Ptpn11* itself (Supplementary Fig. 16). These data indicate broad implications of this unique signaling nexus. In addition to AR and SHP2, ACK1 promotes the phosphorylation and nuclear accumulation of STAT1/3[75–77]. SHP2 and PTP1B dephosphorylates STATs in multiple cancer cells, including prostate cancer, causing downregulation of target genes[78,79]. Whether pSTAT could recruit SHP2 to chromatin to erase pY54-H3 repressive marks to activate a distinct set of genes remains to be seen.

Overall, the evolutionary conservation (Tyr54 site appears to be conserved in all eukaryotes that we checked, including in budding yeast) and differential execution of the pY54-H3 epigenetic program in developmental disorders and cancer reveal the fundamental importance of this primordial signaling nexus. Further, ACK1/SHP2/pY54-H3/AR signaling balances the loss- and gain-of-function mechanism of the pathogenesis of SHP2-associated human diseases, and thus opens previously untested therapeutic modalities. This report demonstrates

the epigenetic activity of the PTP superfamily of enzymes and reveals a divergence in PTP superfamily members based on distinct chromatin modifying function. Further, it reveals pY54-H3 epigenetic mark is a 'sensor' of the AR levels, and crucial regulator of the development of male reproductive system.

## Methods

All the experiments for the study were performed following standards according to the protocol written and approved by the Institutional Biological and Chemical (IBC) Safety Committee, Washington University in St. Louis (IBC protocol no. 7100).

### Cell lines

RWPE-1, HEK293T, LNCaP, VCaP, PC3, LNCaP-C4-2B and DU145 cells were obtained from ATCC. LNCaP, PC3 and DU145 prostate cancer cell lines were grown in RPMI-1640 supplemented with 10% FBS (Invitrogen). HEK293T and VCaP cells were grown in DMEM with 10% FBS (Invitrogen). RWPE-1 (ATCC) was grown in Keratinocyte media with L-glutamine (Invitrogen) supplemented with 2.5 mg EGF (Invitrogen) and 25 mg Bovine Pituitary Extract (Invitrogen). LNCaP-C4-2B and LAPC4 cells were grown[28]. All cultures were maintained with 50 units/ml of penicillin/streptomycin (Invitrogen) and cultured in 5% CO2 incubator. All cultures were tested for mycoplasma contamination every 2 months using the PCR Mycoplasma Test Kit I/C (PromoKine). All cell lines tested negative for mycoplasma contamination. Identities of all cell lines were confirmed by Short Tandem Repeat (STR) Profiling.

### Mice studies

All animal experiments were performed using the standards for humane care in accordance with the NIH Guide for the Care and Use of Laboratory Animals. Mice breeding and colony maintenance was performed according to IACUC protocols approved in writing by Washington University in St. Louis, Department of Comparative Medicine (IACUC protocol no. 20180259). All mice were co-housed with 3–5 mice per cage and maintained in a controlled pathogen-free/germ-free environment with a temperature of 20-23 °C, 12/12 h light/dark cycle, 50–60% humidity, and food and water provided ad libitum. For all mice experiments, tumor volumes of 1500mm³ or end of treatment period, was considered as the humane end point. The maximum allowed weight loss was 10% of the body weight. The maximum limit for tumor burden and weight loss was not exceeded throughout the study. 5-6 weeks old male SCID mice (CB17/Icr-Prkdcscid/IcrIcoCrl Strain code: 236) were purchased from Charles River Laboratories, USA. For xenograft tumor growth study, $2 \times 10^6$ VCaP cells were suspended in 200 μl of PBS with 50% Matrigel and were implanted subcutaneously into the dorsal flank of castrated six-week-old male SCID mice. Once the tumors reach approximately 100mm³ in size (about 4–5 weeks), mice were administered orally with (*R*)-**9b** resuspended in 6% Captisol (20 mg/Kg), SHP099 resuspended in 6% Captisol, enzalutamide suspension (30 mg/kg) was prepared in corn oil, five times a week. Tumor growth and body weight were checked thrice a week. Tumor volume was calculated using the formula: volume = (width)2 × length/2. Formation of tumors was monitored over a 10-week period. At the end of the experiment (8 weeks post cancer cell injection), the animals were humanely euthanized by carbon dioxide inhalation, followed by cervical dislocation. Tumors were extracted and weighed. The *Ack1* KO mice were developed and described[49]. *Ptpn11*Y279C/+ or NSML/+ Knock in mice were developed and described[61].

### Induced pluripotent stem cells (iPSCs)

Induced pluripotent stem cells (iPSCs) were derived from fibroblasts of the NSML patient (Q510E iPSC), and the healthy individual (WT iPSCs)[80,81] with informed consent approved by the local ethics committee of the medical faculty at the Justus-Liebig-University Giessen,

Germany. A fibroblast sample came from these publications was used to generate the iPSCs. We did not need IRB approvals for these cells.

## Inhibitors or ligand treatment

For all the studies transfected or un-transfected cells were treated with (R)-**9b**, SHP099, enzalutamide or abiraterone (at 1 μM), as per experiments. iPSCs were treated with DHT at concentration 100 nM for 24 h or 1μM for 48 h for in respective experiment. Cells were then utilized for cell lysate preparation, nuclear cytosolic fractionation, binding assays, immunofluorescence, ChIP assays and qRT-PCRs.

## Knockdown of by siRNA Interference

Various prostate cancer cell lines were transfected with test siRNA or control siRNA by using X-tremeGENE siRNA transfection reagent (Sigma-Aldrich, 4476093001). Cells were cultured for 48 h and then harvested for immunoblotting experiments. Following siRNAs were used in the present study: TNK2 siRNA#1 (Dharmacon RNA Technologies, D-003102-13), TNK2 siRNA#2 (Dharmacon RNA Technologies, D-003102-11), AR siRNA (Santa Cruz Biotechnology, sc-29204), PTPN11 (SHP2) siRNA (Thermo fisher scientific, AM16708), PTPN1 siRNA (Horizon Discovery, D-003529-01-0005), PTPN2 siRNA (Horizon Discovery, D-008969-03-0005), PTPN6 (SHP1) siRNA (Horizon Discovery, D-009778-01-0005), NCOR1 siRNA (Horizon Discovery, D-003518-01-0005), NCOR2 siRNA (Horizon Discovery, D-020145-01-0005) and control siRNA-A (Santa Cruz Biotechnology, sc-37007).

## Generation and affinity purification of the p-Y-54 H3 antibody

pY54-H3 peptide was coupled to immunogenic carrier proteins and the pY54-H3 antibodies were custom synthesized by 21st Century Biochemicals.

## pY54-H3 peptide: C-Ahr-ALREIRR(pY)QKSTEL-Amide

Briefly, two rabbits were immunized twice with the phosphopeptide, and the sera from these rabbits was affinity purified. Two antigen-affinity columns were used to purify the phospho-specific antibodies. The first column was the nonphosphopeptide affinity column. Antibodies recognizing the unphosphorylated residues of the peptide bound to the column. The flow-through fraction was collected and then applied to the second column, the phosphopeptide column. Antibodies recognizing the phospho-Tyr residue bound to the column and were eluted as phospho-specific antibodies.

## Western blot analysis

RWPE-1, VCaP, C4-2B, LAPC-4 and HEK293T cells were transfected and/or treated as per experimental requirement and were harvested and lysed by sonication in RLB[35]. Lysates were quantitated and 50 μg, unless specifically mentioned, of protein lysates were fractionated by SDS-PAGE, and transferred onto a PVDF membrane. After blocking in 3% bovine serum albumin (BSA), membranes were incubated with the following primary antibodies: anti-pY-SHP2 (Cell Signaling Technology, Cat#5431 S, Clone D66F10, 1:1000), anti-pY54-H3 (21st Century, this paper, 1:1000), anti-H3 (Cell Signaling Technology, Cat# 14269, Clone 1B1B2, 1:1000), anti-SHP2 (Cell Signaling Technology, Cat#3397 S, Clone D50F2, 1:1000), anti-FLAG (Cell Signaling Technology, Cat# 14793 S, Clone D6W5B, 1:1000), anti-HA (Cell Signaling Technology, Cat# 2367 S, Clone 6E2, 1:1000), anti-ACK1 (Santa Cruz Biotechnology, Cat# sc-28336, Clone A11, 1:1000), anti-p-Y-284-TNK2/ACK1 (Millipore Sigma, Cat#09-142, 1:1000), anti-His (Cell Signaling Technology, Cat# 2366, Clone 27E8, 1:1000), anti-NCOR1 (Cell Signaling Technology, Cat# 5948 S, 1:1000), anti-SMRT (NCOR2) (Cell Signaling Technology, Cat# 62370, 1:1000), anti- phosphoTyrosine (pTyr) (Santa Cruz Biotechnology, Cat# SC-508, Clone PY20, 1:500), anti-IR (Cell Signaling Technology, Cat# 3025, Clone 4B8, 1:1000), anti-HER4 (Cell Signaling Technology, Cat# 4795, Clone 111B2, 1:1000), anti-SRC (Cell Signaling Technology, Cat# 2108 S, 1:1000), anti-MYC (Cell

Signaling Technology, Cat# 2276 S, Clone 9B11, 1:1000), anti-HSP90 (Cell Signaling Technology, Cat# 4874, 1:1000), anti-PTPN1 (Santa Cruz Biotechnology, Cat# sc-133259, Clone D-4, 1:1000), anti-PTPN2 (Santa Cruz Biotechnology, Cat# sc-376864, Clone E11, 1:1000), anti-SHP1 (Santa Cruz Biotechnology, Cat# sc-7289, Clone D11, 1:1000) and anti-Actin (Sigma, Cat# A2228, Clone AC-74, 1:9000). Following three washes in PBS-T, the blots were incubated with horseradish peroxidase-conjugated secondary antibody. The blots were washed thrice, and the signals visualized by Pierce™ ECL Western Blotting Substrate according to manufacturer's instructions (Thermo Fisher Scientific, 32209).

For immunoprecipitation studies, cells or xenograft tumors were lysed by sonication in RLB, the lysates were quantitated and 0.5 to 1 mg of protein lysate was utilized for immunoprecipitation using 1–2 μg of respective antibody coupled with protein A/G sepharose (Santa Cruz Biotechnology, sc-2003)[37]. Samples were then washes with RLB and PBS buffers. The beads were boiled in sample buffer and immunoblotting was performed as described above. Densitometric analysis using ImageJ software (ImageJ, NIH, USA) was performed for representative immunoblot images after normalizing with actin and fold change intensity for each lane is incorporated, wherever required.

## Nuclear and cytosolic extract preparation

Prostate cancer cells were harvested as per various experimental conditions and cell pellets were utilized for nuclear and cytosolic extract preparation. Briefly, $3.2 \times 10^6$ cells were resuspended in 300 μl of first buffer composed of 10 mM HEPES, 1.5 mM MgCl$_2$, 10 mM KCl, 0.5 mM DTT, 0.05% NP40 at pH 7.9 and centrifuge at 845 x g for 10 min at 4 °C. Supernatant was removed as cytosolic extract. Remaining pellet was resuspended in 500 ul of second buffer containing 5 mM HEPES, 1.5 mM MgCl2, 0.2 mM EDTA, 0.5 mM DTT, 26% glycerol (v/v) at pH 7.9 and then was centrifuged at 845 x g for 2 min at 4 °C. Remaining pellet was resuspended in 200 μl of second buffer and homogenize on ice till clear suspension (10–12 pulses). Samples were then incubated on ice for further 30 min and then centrifuged for 20 min at 4 °C. Total cell lysate (CL) is used as control.

## In vitro kinase assay

To determine ACK1 catalytic activity to remove pY54-H3 marks by recruiting Shp2, in vitro kinase assay was performed. VCaP cells were transfected with HA tagged-ACK1, kdACK1, Flag tagged-Shp2 or vector. Post 48 h of transfection, cells were treated with (R)-**9b** or SHP099 or DMSO for 12 hr. Cell lysates were prepared and incubated with H3 peptide-bead conjugate for 3 hr in kinase assay buffer[37]. The reaction was separated on SDS-PAGE, followed by immunoblotting with pY54-H3 antibody to detect specific phosphorylation on H3. Inputs were also subjected to western blotting to detect ACK1, SHP2 and Actin levels.

Nucleosomes (Active Motif) were incubated with purified ACK1[28], or purified SHP2, or ACK1 followed by SHP2 incubation for 60 min at 37 °C. The reaction was separated on SDS-PAGE followed by immunoblotting with pY54-H3, total H2B, H2A, H3 and H4 antibodies.

## Pull down/binding assays

Two human histone H3 peptides spanning amino acids 44-59 were synthesized with Tyr54 at middle of the peptide. The sequences are as follows:

**pY54-H3 (44-59):** VALREIRR**pY**QKSTE
**H3 (44-59):** VALREIRRYQKSTE

Both the peptides were biotinylated at C-terminus and immobilized on streptavidin-sepharose beads.

For NCORs binding assay, the beads were incubated with cell lysates made from VCaP cells treated with (R)-**9b**/SHP099 and resolved on SDS-PAGE for immunoblot based detection of NCORs. pAKT peptide was used as non-specific control for the reaction. To confirm interaction between SANT domain and pY54-H3, in vitro binding assay

was performed with biotinylated pY54-H3 or H3 peptides that were incubated and with varying concentration of His-tagged SANT domain (SD) peptide. Samples were then subjected to immunoblotting for detection of His-tagged peptide.

## Peptide synthesis

Synthetic peptides (including biotin-conjugated peptides) were custom synthesized at GenScript, NJ. Peptides were dissolved in sterile water and aliquots were stored in −20 °C.

## SHP2 phosphatase assay

For SHP2 phosphatase assay, pY54-H3 peptide and non-phosphorylated H3 peptides were incubated with purified SHP2 for 60 min at 37 °C. The reaction was spotted on nitrocellulose membrane, followed by immunoblotting with pY54-H3 antibody. SHP2 mediated dephosphorylation of pY54-H3 was also checked by incubation with control pY37-H2B, pY88-H4, ac130-H2A and pY363-AR peptides, for 60 min at 37 °C. The reaction was spotted on a nitrocellulose membrane, followed by followed by immunoblotting with respective antibodies.

## Mass spectrometry

Cells were lysed in denaturing buffer containing 8 M urea, 20 mM HEPES (pH 8), 1 mM sodium orthovanadate, 2.5 mM sodium pyrophosphate and 1 mM β-glycerophosphate. Bradford assays determined the protein concentration for each sample. Protein disulfides were reduced with 4.5 mM DTT at 60 °C for 30 minutes and then cysteines were alkylated with 10 mM iodoacetamide for 20 minutes in the dark at room temperature. Trypsin digestion was carried out at room temperature overnight with enzyme to substrate ratio of 1:20, and tryptic peptides were acidified with aqueous 1% trifluoroacetic acid (TFA) and desalted with C18 Sep-Pak cartridges according to the manufacturer's procedure. Following lyophilization, peptide pellets were re-dissolved in immunoaffinity purification (IAP) buffer containing 50 mM MOPS pH 7.2, 10 mM sodium phosphate and 50 mM NaCl. Phosphotyrosine-containing peptides (pY) were immunoprecipitated with p-Tyr-1000 beads (Cell Signaling Technology #8803 S).

A nanoflow ultra-high-performance liquid chromatograph and nanoelectrospray orbitrap mass spectrometer (RSLCnano and Q Exactive plus, Thermo) were used for LC-MS/MS. The sample was loaded onto a pre-column (C18 PepMap100, 2 cm length x 100 μm ID packed with C18 reversed-phase resin, 5 μm particle size, 100 Å pore size) and washed for 8 minutes with aqueous 2% acetonitrile and 0.1% formic acid. Trapped peptides were eluted onto the analytical column, (C18 PepMap100, 25 cm length x 75 μm ID, 2 μm particle size, 100 Å pore size, Thermo). A 120-minute gradient was programmed as: 95% solvent A (aqueous 2% acetonitrile + 0.1% formic acid) for 8 minutes, solvent B (aqueous 90% acetonitrile + 0.1% formic acid) from 5% to 38.5% in 90 minutes, then solvent B from 50% to 90% B in 7 minutes and held at 90% for 5 minutes, followed by solvent B from 90% to 5% in 1 minute and re-equilibration for 10 minutes using a flow rate of 300 nl/min. Spray voltage was 1900 V. Capillary temperature was 275 °C. S lens RF level was set at 40. Top 16 tandem mass spectra were collected in a data-dependent manner. The resolution for MS and MS/MS were set at 70,000 and 17,500 respectively. Dynamic exclusion was 15 seconds for previously sampled peaks.

Data Analysis: MaxQuant (version 1.2.2.5) was used to identify peptides using the UniProt human database and quantify the MS1 precursor intensities. Up to 2 missed trypsin cleavages were allowed. The mass tolerance was 20 ppm first search and 4.5 ppm main search. Carbamidomethyl cysteine was set as fixed modification. Phosphorylation on Serine/Threonine/Tyrosine and Methionine oxidation were set as variable modifications. Both peptide spectral match (PSM) and protein false discovery rate (FDR) were set at 0.05. Match between runs feature was activated to carry identifications across samples. For data upload to PRIDE/ProteomeXchange, similar database searches were performed with Mascot (www.matrixscience.com) in Proteome Discoverer (Thermo).

## Chromatin immunoprecipitation (ChIP) and ChIP-sequencing

Prostate cancer cells ($5\times10^7$ cells) were either untreated or treated with (R)-**9b** or SHP099 were harvested, and lysates were prepared. Briefly, cell pellets were resuspended in RLB buffer and sonicated for 12 seconds on ice for 3 times. Similarly, prostates of WT and *Ack1* KO mice were homogenized and processed. This suspension containing soluble chromatin was incubated with pY54-H3 or IgG antibodies and protein-G/A magnetic beads at 4 °C. A part of suspension soluble chromatin without immunoprecipitation was saved as input DNA. The amount of immunoprecipitated DNA was determined by real-time PCR. The complexes were washed with ChIP buffer 1 and 2 (Active Motif), eluted with elution buffer and subjected to proteinase-K treatment. Fixed ChIP DNA was 'reversed' cross linked and subjected to proteinase-K treatment. ChIP DNA was purified using PCR DNA purification column.

Fifty nanograms of immunoprecipitated DNA was fragmented to 300 base pairs using a Covaris M220 Focused-ultrasonicator (Covaris, Inc., Woburn, MA) and then used to generate sequencing libraries using the Kapa Hyper Prep Kit (Kapa Biosystems, Wilmington, MA). The size and quality of the library was evaluated using the Agilent BioAnalzyer, and the library was quantitated with the Kapa Library Quantification Kit. Each enriched DNA library was then sequenced on an Illumina NextSeq 500 sequencer. The sequencing yield was very good, with over 90 million reads in each sample, of which 82.8 (IGF) and 77.9 million (IgG), respectively, mapped uniquely to the human genome GRCh37.

## Immunofluorescence

Prostate cancer cells ($3\times10^2$) under various experimental conditions were seeded on cover slips and allowed to adhere for 24 h at 37 °C. Thereafter, cells were treated with inhibitors as per experimental conditions. Cell were then processed and permeabilized[82]. Next, the cells were incubated with the primary antibodies, pY-SHP2, pY54-H3, FLAG (SHP2), HA (ACK1), ACK1 and AR at a dilution of 1:500 in blocking buffer for 1 h. The excess antibody was washed off with 1X PBS. The cells were incubated with Alexaflour 488 conjugated anti-rabbit secondary antibody (1:1000 dilution) and Alexaflour 594 conjugated anti-mouse secondary antibody (1:1000 dilution) for 30 min. The cells were washed, and coverslips were mounted on glass slides in DAPI (Thermo Fisher Scientific, D1306) containing mounting media. The cells were visualized using Zeiss LSM 880 II Airyscan FAST Confocal Microscope with a 63X oil immersion objective. Zen blue software was used to acquire and process images.

## Immunohistochemical staining

Fine sections (5 μm) were prepared from formalin fixed paraffin embedded tumor tissues and fixed on glass slides. For immunohistochemistry, slides were deparaffinization and antigen retrieval was performed[45]. Horse serum (2.5%) was used for blocking for 1 h. After washing with PBST, slides were probed with pY-Shp2 (1:500) and pY54-H3 (1:100) and incubated at 4 °C overnight. Slides were washed with PBST and incubated with biotinylated horse anti-rabbit IgG. The sections were counterstained with hematoxylin (Vector Laboratories) after developing with peroxidase substrate[37]. Tissue sections were then examined, and images were captured with a brightfield microscope (EVOS M5000 Imaging System, Invitrogen) at 20x magnification. Histopathological analysis was performed for each tissue section and nuclear scores (ns) were marked for IHC staining for both pY-Shp2 and pY54-H3.

## Flow cytometry

Increase in pY54-H3 upon *ACK1* knock down was assessed by flow cytometry. Under sterile conditions, femur, heart, liver, and prostate

were harvested from naïve WT and *Ack1* KO mice. Single cells were made, and RBCs were lysed using RBC lysis buffer. $1 \times 10^6$ cells were incubated with Live/Dead Aqua (Invitrogen, Cat#L34957, 1:400) and fixed. Cells were then permeabilized and incubated with pY54-H3 (1:300). Flow cytometry was performed after intracellular staining with anti-rabbit Alexa Fluor® 488 antibody (Invitrogen, Cat# A11034, 1:700).

### Quantitative RT-PCR and ChIP-qPCR

Cells under various experimental conditions were used for RNA isolation and cDNA preparation[49]. All RT reactions were done at the same time so that the same reactions could be used for all gene studies. For the construction of standard curves, serial dilutions of pooled sample RNA were used (50, 10, 2, 0.4, 0.08, and 0.016 ng) per reverse transcriptase reaction. One "no RNA" control and one "no Reverse Transcriptase" control were included for the standard curve. Three reactions were performed for each sample: 10 ng, 0.8 ng, and a NoRT (10 ng) control. Real-time quantitative PCR analyzes were performed using the ABI PRISM 7900HT Sequence Detection System (Applied Biosystems). All standards, the no template control ($H_2O$), the No RNA control, the no Reverse Transcriptase control, and the no amplification control (Bluescript plasmid) were tested in six wells per gene (2 wells/plate x 3 plates/gene). All samples were tested in triplicate wells each for the 10 ng and 0.8 ng concentrations. PCR was carried out with SYBR® Premix Ex TaqTM II TB green premix (TaKaRa, RR82LR) using 2 µl of cDNA and the primers in a 20 µl final reaction mixture. After 2 min incubation at 50 °C, the reaction was activated by 10 min incubation at 95 °C, followed by 40 PCR cycles consisting of 15 s of denaturation at 95 °C and hybridization of primers for 1 min at 55 °C. Dissociation curves were generated for each plate to verify the integrity of the primers. Data were analyzed using StepOne and StepOnePlus software version 2.3 and exported into an Excel spreadsheet. The actin or GAPDH data were used for normalizing the gene values; i.e., ng gene/ng Actin or GAPDH per well. The primer sequences are shown in Supplementary Table 2.

### Computational analysis of ChIP-Seq data: Sequence analysis

The 75-nt paired-end sequence reads were mapped to the genome using the BWA-MEM algorithm. Alignment information for each read was stored in the output file *.bam. Only reads that mapped uniquely with proper pairing were used in the subsequent analysis.

### Determination of fragment density

Because the 5′ ends of the sequence reads represent the end of the ChIP or immunoprecipitation fragments, the reads were extended in silico (using MAC2) at their 3′ ends to a length of 173-244 bp, based on the fragment length calculated from the read pairs.

### Peak finding

Peak regions were called using the MACS2 software with the following options -f BAMPE -SPMR -q 0.01 -broad. The "BAMPE" option was used for calculating fragment lengths from the paired end reads, "SPMR" for normalizing read depths to number of fragments per million reads, "broad" for compositing broad regions from nearby peak regions, and the qvalue (FDR) cutoff was set to 0.01.

### Motif analysis

Using the HOMER (v4.7, 8-25-2014) program (http://homer.salk.edu/homer/) we found 13 de novo motifs in LNCaP cells treated with (*R*)-**9b**, pY54-H3 ChIP data, which are shown in Fig. S6a. We also identified 3 known motifs, in Ack1 KO prostates, shown in Fig. S6b.

### Tissue Microarray (TMA) Analysis

The prostate TMAs were obtained from US Biomax (PR807c: 80 cases) for our study for which we are exempt from IRB approval, as no personal information about patients is sought. The tissue array slides (including positive and negative controls) were stained for the antibodies. The slides were dewaxed by heating at 65° Celsius for 60 min and washed two times, 15 minutes each, with xylene. Tissues were rehydrated by two series of 10 min washes in 100%, 95%, and 70% ethanol and distilled water. Antigen retrieval was performed by heating the samples at 95 °C for 25 min in 10 mmol/L sodium citrate (pH 6.0). The slides were cooled in PBS for 30 min, with 10 min changes of PBS, and permeabilized using 0.2% Triton-X100 in PBS for 10 min. Slides were washed with PBS + 0.2% Tween-20 for 10 min. After blocking with universal blocking serum (DAKO Diagnostic, Mississauga, Ontario, Canada) for 30 min, the samples were incubated with rabbit monoclonal pY580-SHP2 (1:300 dilution) and rabbit polyclonal pY54-H3 antibody (1:300 dilution) at 4 °C overnight. The sections were incubated with biotin-labeled secondary and streptavidinperoxidase for 30 min each (DAKO Diagnostic). The samples were developed with 3,39-diaminobenzidine substrate (Vector Laboratories, Burlington, Ontario, Canada) and counterstained with hematoxylin. Following standard procedures, the slides were dehydrated and sealed with cover slips. The pY-SHP2 and pY54-H3 staining were examined in a blinded fashion by pathologist (C.W.). The positive reactions were scored into four grades according to the intensity of staining: 0, 1 + , 2+ and 3 + .

### Statistical analysis

Data for all experiments were analyzed with Prism software (GraphPad). All statistical analyzes were performed using Student's *t*-test or one-way ANOVA unless otherwise specified. Boxplots were used to summarize the intensity distribution at each progression stage. Furthermore, the Tukey–Kramer method was performed to examine between which pairs of stages the expression levels are different. This post-hoc procedure adjusts for all pairwise comparisons and simultaneous inference. When more than one sample was obtained from a patient, the intensity of the most progressed stage was used for the analysis. Correlation between pY54-H3 and Shp2 and pY-ACK1 was explored using Spearman ranked correlation analysis. Statistical differences between the groups were determined using log-rank test. *p* values less than 0.05 were considered statistically significant.

### Reporting summary

Further information on research design is available in the Nature Portfolio Reporting Summary linked to this article.

## Data availability

The mass spec data is submitted in ProteomeXchange (PRIDE) with the accession number PXD037546. ChIP-sequencing datasets generated in this study have been deposited in GEO database with the GEO accession number: GSE214870. The remaining data are available within the Article, Supplementary Information or Source Data file. Source data are provided with this paper.

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

## Acknowledgements

We thank Dr. Benjamin Neel for the critical reading of the manuscript and suggestions. We also thank our lab member, Dr. Elliot Bradshaw, for manuscript reading. N.P.M. is a recipient of NIH/NCI grants (5R01CA227025, 1R01CA273054 and 1R01CA276502), and Department of Defense grant (W81XWH-21-1-0202). N.P.M. also acknowledges Hamacher Family Prostate Cancer Research fund. K.M. is a recipient of the Department of Defense award W81XWH-21-1-0203. A.Y. is supported by grant EJP RD COFUND-EJP no. 825575. M.I.K. is supported by NIH grants (R01-HL102368 and R01CA190838), the Department of Defense (W81XWH-21-1-0784), American Heart Association (20TPA35490426), the Masonic Medical Research Institute support and funding from Onconova Therapeutics. This work has been supported in part by the Proteomics and Metabolomics Core Facility at the Moffitt Cancer Center (P30-CA076292).

## Author contributions

S.C. and N.P.M conceived the study; S.C., D.S., and N.P.M designed the experiments; S.C. and D.S. performed the experiments; K.M. generated *Ack1* KO mice, C.W. read slides; J.L. performed statistical analysis; M.C.H., S.Le.S. and M.I.K. bred *Ptpn11* knockin mice and provided organs. L.N.S. grew and provided Q510E iPSCs; A.Y. provided *Ptpn11* knockin mice organs and discussion of the data; C-K.Q. provided *Ptpn11^{E76K/+}Nestin*-Cre^+ mice organs; T.L. analyzed ChIPseq data; J.K. and B.F. performed mass spectrometry studies; M.S. performed crystal structure-based ACK1-SHP2 interaction analysis; S.C., D.S. and N.P.M. wrote the manuscript. All authors read the manuscript.

## Competing interests

Patents "Inhibitors of ACK1/TNK2 Tyrosine Kinase" (patent no. 9,850,216, 10,017,478 and 10,336,734) covers (*R*)-**9b** compound. N.P.M. and K.M. are named as inventors. TechnoGenesys has licensed these patents. N.P.M. and K.M. are cofounders of TechnoGenesys Inc., own stocks, and serve as consultant. The remaining authors declare no competing interests.
