## [Peer Review File · Nature Communications]

SHP2 as a Primordial Epigenetic Enzyme Expunges Histone H3 pTyr-54 to Amend Androgen Receptor HomeostasisREVIEWER COMMENTS

Reviewer #1 (Remarks to the Author):

Chouhan and colleagues analyzed how a newly discovered interplay between the tyrosine kinase ACK1 and the phosphatase SHP2 controls epigenetic signaling. They uncovered that SHP2 is a distinct class of an epigenetic enzyme that specifically erases the pY54-H3 repressive marks. This mechanism controls a novel ACK1/SHP2/pY54-H3/AR signaling node with importance for human diseases. Their mechanistic insights provide a fresh link between cytoplasmic signaling and nuclear, epigenetic processes. This work is very appealing and interesting at various biological levels. As always with works from the Mahajan lab, a very large number of experiments that were collected with numerous techniques and systems are presented. I strongly support publication of the work, with some revisions. These are surely doable by Chouhan and colleagues.

My suggestions to revise and improve the manuscript are:

Figure 1 demonstrates that SHP2 erases pY54-H3 epigenetic marks upon its activation by ACK1 in cell lines and tissues from mice. The data were collected with pharmacological and genetic approaches.

Comments:

Why was SHP1 not considered as alternative phosphatase? RNAi against SHP1 could show if the observed effects are specific for SHP2 or also exerted by SHP1.

Writing for mammalian proteins is inconsistent. It should be capitalized throughout the manuscript - SHP2 instead of Shp2.

Figure 2 shows that the activity and localization of SHP2 rely on AR, giving a link to prostate cancer biology.

Comment: I suggest showing the total levels of the phosphorylated proteins and to prove downregulation by the siRNAs in all immunoblots.

Figure 3 demonstrates that SHP2 erases pY54-H3 epigenetic marks at the AR gene locus. This data provides new evidence on cytoplasmic-to-nuclear signaling.

Comment: the figure is clear to me. I have no comments.

Figure 4 shows that enhanced pY54-H3 epigenetic marks suppress AR and PSA transcription.

Comment: the figure is clear to me. I have no comments.

Figure 5 demonstrates that the epigenetic readers NCOR1/NCOR2 readout the histone H3 mark that is controlled by ACK1 and SHP2.

Figure 5: NCOR1 & 2 are... This is not wrong but for a scientific paper, I suggest NCOR1 and NCOR2 are... Moreover, the writing should be NCoR1/NCoR2 or all letters in capital NCOR1/NCOR2 (which I suggest having this consistent with SMRT, all capital letters) throughout the manuscript. Does downregulation of NCOR1/NCOR2 by RNAi alter pY54-H3?

Figure 6 shows depletion of pY54-H3 marks and increased pSHP2 levels during prostate cancer progression to malignant stage.

Comment: the figure is clear to me. I have no comments. Very fascinating data, just two short questions – how frequent are mutations of ACK1 and SHP2 in such tumor cells? Does DepMap show evidence for ACK1 and SHP2 as dependency factors?

Figure 7 illustrates that an inhibition of ACK1, SHP2, or AR suppresses AR-dependent transcription and prostate tumor growth. This is linked to altered pY54-H3 marks.

Comment: I suggest showing the total levels of the phosphorylated proteins in the immunoblot (Fig. 7D).

Figure 8 shows a linkage between pY54-H3 marks and NSML.

Comments:

Total histone H3 levels should be provided.

I suggest incorporating Fig. 9 into Fig. 8.

Discussion:

The manuscript is well written, but the last paragraph of the Discussion should be revised. In addition to JAKs, ACK1 phosphorylates STAT1/STAT3. SHP2 and TCP45 remove this posttranslational modification in various cancer cells, including those from prostate cancer; see e.g., PMID: 28739485, PMID: 19171783, PMID: 28666623, PMID: 32201212; PMID: 36752816. There is ample evidence on the roles of STAT3 in prostate cancer; see e.g., PMID: 26198641, PMID: 36916439, etc. Could such mechanisms be related to the new findings of the manuscript? how might this be analyzed in future studies?

Minor:

- Ref. 62 Bohmer should be Böhmer
- Better use and instead of &

Reviewer #2 (Remarks to the Author):

In this article, the authors have identified new modification on histone H3 (pY54-H3). They proposed ACK1 controlled SHP2 as a possible phosphatase that regulates this modification. They showed that SHP2 phosphorylation by ACK1 enhances its phosphatase activity, and promotes its interaction with AR. Specifically, they showed that SHP-2 promotes the expression of androgen receptor and its target genes by dephosphorylating tyrosine 54 residue of histone H3 in the nucleus. Moreover, the authors demonstrated that prostate tumors in mice have high levels of pSHP2 and pACK1 activity and low levels of pY54-H3 levels and higher AR expression. On the other hand, NSML patients with compromised SHP1 activity display higher pY54-H3 levels that may contribute to AR dependent phenotypes in these patients.

1. While the identification of a new histone modification is interesting, the specificity of phosphatase is a major concern. The pY54-H3 is changing with all the inhibitors used in this paper. For example, pY54-H3 increased in response to inhibitors like SF1670, BCI, Salubrinal, KY226, SSBG and SHP099 which target different phosphatases. The observed changes in pY54-H3 phosphorylation may also be a result of off-target effects of the inhibitors or nonspecific binding to other proteins.
2. It is not clear from the experiments whether H3-Y54 dephosphorylation is indeed a direct dephosphorylation target of SHP2 or an indirect effect. An assay that shows SHP2 activity directly on this site is missing.
3. Another major concern in the paper: Given that the modification is in H3, a core histone that might affect chromatin related events globally, any alterations in such modification might lead to global changes in transcription. However, all the data limited to changes in single loci of AR upon Ack1 or SHP2 need a relook. This raises a question on the physiological relevance of this modification.
4. Is this modification limited to AR dependent cells or other cell types as well? Under what conditions is this modification added and/or removed.
5. No data is shown to demonstrate if Y54 phosphorylation/dephosphorylation is occurring with the nucleosome associated Histones or free histones.
6. What is the status of the nucleosomes if you express Y54 phospho mimetic? Does Y54 mutant affect binding with other histones and whether it incorporate into the nucleosomes properly and/or affect DNA binding? This information is quite important to clearly assess the role of Y54 modification.
7. Majority of the data that show changes in Y54 modification is done with overexpression experiments or with inhibitors which may not be always specific.

Therefore, with these concerns, the manuscript may not be suitable for publication at Nature Communications.

Other Comments:

1. A rationale for testing only SHP2 (phosphatase) among many kinases in Fig S1E is not clear.

Figure S1E- expression of kinases/SHP-2 not shown.

2. Inhibitor treatment period varies across the figures. (For example, in Figure 1 cells were treated for 8 hours. The same cell type in Figure 5 and Supplementary Fig2 were subjected to treatment for 18 or 16 hours.

3. What is the effect of inhibitor treatment on the total protein level of SHP2 and ACK1?

4. Figure 1F: SHP2 activity in these tissues is important to be shown.

5. If AR is shuttling SHP2, it is not clear on why SHP2 activity is required for its nuclear translocation (Fig 2C).

6. In Figure 2B it will be better to run cytoplasmic and nuclear extracts in the same SDS PAGE gel to claim nucleocytoplasmic shuttling of pSHP2.

7. Supplementary Fig S2C: If Ack1 is upstream of SHP2 activity as per the hypothesis, why will SHP2 inhibition lead to reduction in Ack1 phosphorylation. This raises a question on the specificity of the inhibitors.

8. What is the difference between supplementary figures 4A and 4E?

9. Supplementary Fig. S12A- figure legend says ChIP was performed using NCoR1 & NCoR2 (SMRT) antibodies but data shown only for NCoR1 in VCaP cells.

10. pSHP-2 antibody detects which phosphorylation?

11. In Supplementary Fig 4D pY54H3 signal is in the cytoplasm (Overexpression of SHP-2 279 mutant).

12. Does SHP2 binds directly to chromatin? Or it remains bound to AR and regulates H3 phosphorylation?

13. Supplementary data S3A-C: Mass spec data for all the identified proteins (not just SHP2) need to be shown in the manuscript as a list may be.

14. Is Ack1 shown to directly phosphorylate Y279 and Y580 sites on SHP2?

15. The authors mention in the discussion that the direct chromatin-modifying role for protein tyrosine phosphatases (PTPs) is not known and this is the first report of the epigenetic activity among the PTP superfamily of enzymes. This may not be entirely true. The authors may check the literature carefully and modify this statement.

16. "PTPN2 (TCPTP), PTPN6 (SHP1) and PTPN22 (PEP) have strong presence in nuclei of immune cells" This statement is factually incorrect. None of the references cited for the same does support it.

17. Most of the western blot labelling is confusing and some figures have labelling mistakes.

Reviewer #3 (Remarks to the Author):

This manuscript by Chouhan et al, describes a novel role for the protein tyrosine phosphatase Shp2 in epigenetic regulation. The authors provide intriguing evidence to support the model that Shp2 through the actions of the kinase, ACK1/TNK2 is trafficked to the nucleus via the androgen receptor (AR) where it "erases" tyrosyl phosphorylated Histone3 at residue 54. The strength of the model is on the novel relationship between Shp2 and the effects of modulation pY54-H3 marks as an epigenetic signal. However, the paper is constructed in a confusing manner because it jumps between multiple cell lines, the Noonan syndrome with multiple lentigines (NSML) model and prostate cancer models to substantiate the clinical importance of the findings. There are additional mechanistic shortcomings in the manuscript that need to be addressed. Most notably more convincing evidence that pY54-H3 is a bona fide Shp2 substrate is required if indeed that is the stated mechanism that is being put forth. Other interpretations of the data are conceivable. Overall, this is a very interesting study however, in its current form additional experiments are required to directly support the proposed model.

Major Comments

1. The authors should provide some rationale for the selection of Shp2 for its ability to modulate H3 tyrosyl phosphorylation. There are other nuclear resident PTPs (e.g. TC-PTP) that would likely serve as equally relevant to test in this model. It seems out of the blue as to how they arrived at Shp2 – what about Shp1?

2. It is unclear how Shp2 is regulated in this model. Shp2 exists in a closed conformation and in this state the basal activity of the phosphatase is typically less than 5% of its maximal. In order for Shp2 to be activated engagement of the SH2 domains are required in all instances. Phosphorylation of Shp2 has only minor modulatory effects on overall catalytic activity and this is largely consistent with the downstream signaling effects. The authors refer to the Jenkins et al work and state that "ACK1 phosphorylates SHP2 leading to activation of its Tyr phosphatase activity". Unfortunately, Shp2 tyrosyl phosphorylation at either Y542 or Y580 as it appears to have been used here is not a proxy (and at least not a reliable one) for intrinsic Shp2 catalysis. Indeed, Shp2 is fully regulable and auto-inhibited through its SH2/PTP domain interactions in the absence of its C-terminus that contains these sites (Hof et al, Cell 1998 Vol. 92 Pages 441-450). Furthermore, pY580 and pY542 on Shp2 serve as auto-dephosphorylation sites such that inhibition of Shp2 catalysis results in upregulation of either one or both of these sites. How Shp2 is activated in the nucleus is a significant confounder in the interpretation of the model presented in this study.
3. The authors need to substantiate that the effects on pY54-H3 are a direct consequence of Shp2 phosphatase acting on this substrate. It is conceivable that pY580 on Shp2 is acting as a binding site for an adaptor protein and subsequently serving an indirect role to control pY54-H3. As best as this reviewer can tell there is no direct evidence that Shp2 is dephosphorylating pY54-H3.
4. What are the effects of pY54-H3 levels when Shp2 is subjected to an in vitro dephosphorylation assay comparing wild type Shp2 with that of Shp2 Y580F? The authors should use their pY54-H3 (44-59): VALREIRRpYQKSTE peptide as a substrate for in vitro dephosphorylation assays.
5. Can the authors use substrate-trapping mutants of Shp2 to show direct enzyme-substrate complex formation between pY54-H3 that is subsequently competed with vanadate in a complex?
6. If the authors are not ascribing a direct mechanism of Shp2 dephosphorylation ("erases") of pY54-H3 then why is it necessary that Shp2 be carried to the nucleus?
7. If Shp2 needs to go to the nucleus as a direct mechanism of pY54-H3 dephosphorylation then does abrogating Shp2/AR complex prevent the dephosphorylation of pY54-H3.
8. The authors cannot conclude as stated in line 175-176 that ACK1 mediated Shp2 phosphorylation is crucial to acquire its optimal phosphatase activity – no phosphatase activity is measured.
9. What is the explanation in Figure 2 for why AR and Shp2 antagonism are required for their interaction. Why do all the inhibitors result in nearly a complete inhibition in the expression levels of AR (see Fig. 2A, panel 5)? If this is correct, it might suggest that the reduced complex formation between Shp2 and AR is mediated by AR. The authors need to address this experimentally.
10. Figure 2A is incomplete, only pY-Shp2 is shown in the nuclear fraction what is the fraction present in the cytosol, if any. What is the distribution between the nuclear and cytosolic pools of total Shp2 this needs to be included in the immunoblot as a control and proper quantitation of these levels performed.
11. It is not clear what is meant in line 206 by "nuclear depletion of Shp2" – is this what the authors really mean. If so, how was this accomplished experimentally since these cells are described as being treated with SHP099.
12. An important control is missing, in Figure 8A, and that is the expression levels of AR in prostate, testis and liver. If their AR levels are different than the wild type in NSML tissues this would confound interpretation of these data given that AR is required for Shp2 to localize to the nucleus.
13. What are the effects of the pY54-H3 status with the Noonan syndrome mutants (e.g. N308D, D61G).

Minor comments:

1. Why is that the authors refer to the actions of Shp2 as an 'eraser" whilst it is understandable that this phrase often used in the epigenetic field, strictly speaking it is jargon. The authors should refer to the actions of the Shp2 on pY54 as dephosphorylation rather than the more ambiguous term of an "eraser". This term is suitable for the conclusion but in the description of the results they need to be clearer.
2. The use of multiple cell lines is often not well justified and made it difficult to understand what the rationale for a particular cell line was being used for (eg LNCaP, C4-2B, VCap,LAPC-4 etc). This makes it hard to ascribe consistency with the model.
3. In the figures the cell type is indicated and in others it is not, can the authors be consistent and show the cell type given that many are used in this study.
4. Figure legends should state the statistical analyses used.
5. The model presented in Figure 9 is a little unclear and should be more precisely generated. For example, it shows the N-SH2 domain of Shp2 binding to phosphorylated (pY) AR. This data was neither demonstrated nor referenced in this study. Why is the AR missing in the NSML patient model?

Point-by-Point Response to Reviewers Comments

Reviewer #1 (Remarks to the Author):

Chouhan and colleagues analyzed how a newly discovered interplay between the tyrosine kinase ACK1 and the phosphatase SHP2 controls epigenetic signaling. They uncovered that SHP2 is a distinct class of an epigenetic enzyme that specifically erases the pY54-H3 repressive marks. This mechanism controls a novel ACK1/SHP2/pY54-H3/AR signaling node with importance for human diseases. Their mechanistic insights provide a fresh link between cytoplasmic signaling and nuclear, epigenetic processes. This work is very appealing and interesting at various biological levels. As always with works from the Mahajan lab, a very large number of experiments that were collected with numerous techniques and systems are presented. I strongly support publication of the work, with some revisions. These are surely doable by Chouhan and colleagues.

My suggestions to revise and improve the manuscript are:

Figure 1 demonstrates that SHP2 erases pY54-H3 epigenetic marks upon its activation by ACK1 in cell lines and tissues from mice. The data were collected with pharmacological and genetic approaches.

Comment: Why was SHP1 not considered as alternative phosphatase? RNAi against SHP1 could show if the observed effects are specific for SHP2 or also exerted by SHP1. Writing for mammalian proteins is inconsistent. It should be capitalized throughout the manuscript - SHP2 instead of Shp2.

Response: First, SHP2 was considered because ACK1 is known to interact with SHP2 (Jenkins et al., *Science Signaling*, 2018). We performed siRNA knockdown of other phosphatases including PTPN1 (PTP1B), PTPN2 (TCPTP), PTPN6 (SHP1) and PTPN11 and observed that all of these caused increase in pY54-He levels with PTPN11 having the most activity, followed by PTPN6 and PTPN1 (**Supplementary Fig. 14e** and **f**). Taken together with the inhibitor studies shown in **Supplementary Fig. 14e**, these data indicates that the spatial and temporal regulation of pY54-H3 deposition may be discrete in PTPN1, PTPN2, PTPN6 and PTPN11, resulting in a distinct transcriptional outcome for these nuclear phosphatases. Follow-up studies could identify a distinct functional properties of these nuclear phosphatases in expunging pY54-H3 marks.

Figure 2 shows that the activity and localization of SHP2 rely on AR, giving a link to prostate cancer biology.

Comment: I suggest showing the total levels of the phosphorylated proteins and to prove downregulation by the siRNAs in all immunoblots.

Response: Levels of total AR, ACK and SHP2 to be shown in Figure 2d.

Figure 3 demonstrates that SHP2 erases pY54-H3 epigenetic marks at the AR gene locus. This data provides new evidence on cytoplasmic-to-nuclear signaling.

Comment: the figure is clear to me. I have no comments.

Response: Thank you.

Figure 4 shows that enhanced pY54-H3 epigenetic marks suppress AR and PSA transcription.

Comment: the figure is clear to me. I have no comments.

Response: Thank you.

Figure 5 demonstrates that the epigenetic readers NCOR1/NCOR2 readout the histone H3 mark that is controlled by ACK1 and SHP2.

Comment: The writing should be NCoR1/NcoR2 or all letters in capital NCOR1/NCOR2 (which I suggest having this consistent with SMRT, all capital letters) throughout the manuscript. Does downregulation of NCOR1/NCOR2 by RNAi alter pY54-H3?

Response: We changed to NCOR1/NCOR2. siRNA mediated knockdown of NCOR1 and NCOR2 was performed, which exhibited modest decrease in pY54-H3 levels, suggesting that pY54-H3 marks deposition are not dependent on NCOR1 and NCOR2 (**Supplementary Fig. S12b**).

Figure 6 shows depletion of pY54-H3 marks and increased pSHP2 levels during prostate cancer progression to malignant stage.

Comment: the figure is clear to me. I have no comments. Very fascinating data, just two short questions – how frequent are mutations of ACK1 and SHP2 in such tumor cells? Does DepMap show evidence for ACK1 and SHP2 as dependency factors?

Response: Thank you. Mutations of ACK1 and SHP2 in tumor cells (cBioportal dataset) shows evidence for ACK1 and SHP2 co-occurrence in multiple malignancies. Please see Supplementary Table S3.

Figure 7 illustrates that an inhibition of ACK1, SHP2, or AR suppresses AR-dependent transcription and prostate tumor growth. This is linked to altered pY54-H3 marks.

Comment: I suggest showing the total levels of the phosphorylated proteins in the immunoblot (Fig. 7D).

Response: Total SHP2 and ACK1 has been added in Figure 7d.

Figure 8 shows a linkage between pY54-H3 marks and NSML.

Comments: Total histone H3 levels should be provided. I suggest incorporating Fig. 9 into Fig. 8.

Response: Total H3 has been added in Fig 8a and f. Further, Figure 9 has been incorporated into Fig. 8, it is now 8h.

Discussion:

The manuscript is well written, but the last paragraph of the Discussion should be revised. In addition to JAKs, ACK1 phosphorylates STAT1/STAT3. SHP2 and TCP45 remove this posttranslational modification in various cancer cells, including those from prostate cancer; see e.g., PMID: 28739485, PMID: 19171783, PMID: 28666623, PMID: 32201212; PMID: 36752816. There is ample evidence on the roles of STAT3 in prostate cancer; see e.g., PMID: 26198641, PMID: 36916439, etc. Could such mechanisms be related to the new findings of the manuscript? how might this be analyzed in future studies?

Response: Thank you for suggestion. We have included this information in the discussion section (second last paragraph).

Minor:

- Ref. 62 Bohmer should be Böhmer

Response: Thank you. We made edits

- Better use `and' instead of &

Response: Agree, fixed it.

Reviewer #2 (Remarks to the Author):

In this article, the authors have identified new modification on histone H3 (pY54-H3). They proposed ACK1 controlled SHP2 as a possible phosphatase that regulates this modification. They showed that SHP2 phosphorylation by ACK1 enhances its phosphatase activity, and promotes its interaction with

AR. Specifically, they showed that SHP-2 promotes the expression of androgen receptor and its target genes by dephosphorylating tyrosine 54 residue of histone H3 in the nucleus. Moreover, the authors demonstrated that prostate tumors in mice have high levels of pSHP2 and pACK1 activity and low levels of pY54-H3 levels and higher AR expression. On the other hand, NSML patients with compromised SHP2 activity display higher pY54-H3 levels that may contribute to AR dependent phenotypes in these patients.

1. While the identification of a new histone modification is interesting, the specificity of phosphatase is a major concern. The pY54-H3 is changing with all the inhibitors used in this paper. For example, pY54-H3 increased in response to inhibitors like SF1670, BCI, Salubrinal, KY226, SSBG and SHP099 which target different phosphatases. The observed changes in pY54-H3 phosphorylation may also be a result of off-target effects of the inhibitors or nonspecific binding to other proteins.

Response: We believe (and we discussed it briefly) that other Tyr-phosphatases also possess the ability to dephosphorylate pY54-H3. We have enclosed a new data wherein siRNA was used to knockdown other phosphatases including PTPN1 (PTP1B), PTPN2 (TCPTP), PTPN6 (SHP1) and PTPN11 and observed that all of these caused increase in pY54-H3 levels with PTPN11 having the most activity, followed by PTPN6 and PTPN1 (**Supplementary Fig. 14e and f**). Taken together with the inhibitor studies shown in **Supplementary Fig. 14d**, these data indicates that the spatial and temporal regulation of pY54-H3 deposition may be discrete in PTPN1, PTPN2, PTPN6 and PTPN11, resulting in a distinct transcriptional outcome for these nuclear phosphatases. Follow-up studies could identify a distinct functional properties of these nuclear phosphatases in expunging pY54-H3 marks.

2. It is not clear from the experiments whether H3-Y54 dephosphorylation is indeed a direct dephosphorylation target of SHP2 or an indirect effect. An assay that shows SHP2 activity directly on this site is missing.

Response: Agree. We have added 2 experiments:

To determine whether pY54-H3 is directly dephosphorylated by SHP2, Myc-tagged SHP2 was purified from HEK cells, which was then incubated with the H3 derived peptide (spanning 49-63 aa) with Tyr54 phosphorylated. It was followed by spotting and immunoblotting with pY54-H3 antibodies. SHP2 erased Y54-phosphorylation (**Supplementary Fig. S1g**). In addition, purified nucleosomes were incubated with purified ACK1 and phosphorylated nucleosomes were then incubated with purified SHP2, followed by immunoblotting. ACK1 enhanced pY54-H3 in nucleosomes (**Supplementary Fig. S1h**, lane 2), which was significantly compromised upon SHP2 incubation (**Supplementary Fig. S1h**, lane 4). Overall these data indicates direct dephosphorylation of pY54-H3 by SHP2.

3. Another major concern in the paper: Given that the modification is in H3, a core histone that might affect chromatin related events globally, any alterations in such modification might lead to global changes in transcription. However, all the data limited to changes in single loci of AR upon Ack1 or SHP2 need a relook. This raises a question on the physiological relevance of this modification.

Response: Again agree completely. Yes, H3 Tyr54-phosphorylation is expected to cause widespread changes in global transcriptional levels. We focused on AR to make the point, but we have assessed many other genes which includes, Ly6e, Akt1, Akt2 and Ptpn11 itself etc. That data is now included in **Supplementary figure 15**.

4. Is this modification limited to AR dependent cells or other cell types as well? Under what conditions is this modification added and/or removed.

Response: AR is expressed in diverse range of human tissues including in bone, muscle, prostate, adipose tissue and the reproductive, cardiovascular, immune, neural and hematopoietic system. Significantly, ACK1 and SHP2 also have widespread expression, which indicates that this pathway

may be operational in multiple tissues. At least in prostate cells, there appears to be dependence on AR; AR inhibitors such as Enzalutamide and Abiraterone were fairly effective in increasing pY54-H3 levels (Fig 2a, compare lanes 1 to 4 and 5, in panel 3).

Currently we are exploring whether tissues lacking AR, but expressing other closely related nuclear hormone receptors such as ER (estrogen receptor), PR (progesterone receptor) or GR (glucocorticoid receptor) possess ability to promote SHP2 nuclear translocation and alter pY54-H3, and if yes, what is the significance of such signaling. However, these are distinct studies and would need separate investigation.

5. No data is shown to demonstrate if Y54 phosphorylation/dephosphorylation is occurring with the nucleosome associated Histones or free histones.

Response: Please see above (point#2).

6. What is the status of the nucleosomes if you express Y54 phospho mimetic? Does Y54 mutant affect binding with other histones and whether it incorporate into the nucleosomes properly and/or affect DNA binding?

Response: Protein phosphorylation is commonly mimicked by the charged amino acids aspartate and glutamate. Although it represents threonine- and serine-phosphorylations, these substitutions inaccurately represent phosphotyrosine when the precise geometries or the aromatic ring of the tyrosine is important. Indeed, to address this issue, nonnatural phosphotyrosine mimetic, *p*-carboxymethyl-l-phenylalanine (pCMF) were used to represent phosphorylated RAD51 protein (*PNAS*, 113, E6045-E6054, 2016). Instead, we have performed studies where we have incubated nucleosomes with purified ACK1 and demonstrated robust Tyr54-phosphorylation (**Supplementary Fig. S1g**, lane 2).

ChIP performed in various cell lines and organs from mice provides evidence that H3 Y54-phosphorylation does not dramatically affects binding of H3 with the other histones, incorporation into the nucleosomes and DNA binding. If that were to be the case, performing ChIP using pY54-H3 itself would have been difficult (indeed impossible).

7. Majority of the data that show changes in Y54 modification is done with overexpression experiments or with inhibitors which may not be always specific.

Response: Well, the biochemical characterization was performed using transfections, inhibitor treatments and siRNA knockdown. However, all these experiments were also further validated using multiple organs of *Ack1* knockout mice (Figure 1f), SHP2 knock-in mice (Figure 8a, c, d, and e). Indeed, we have also validated the data in PBCs from patients (Figure 8f and g). We believe, taken together these data establish the signaling pathway *in vivo*.

Other Comments:

1. A rationale for testing only SHP2 (phosphatase) among many kinases in Fig S1E is not clear. Figure S1E- expression of kinases/SHP-2 not shown.

Response: We have addressed the role of other Tyr-phosphatases, the data is shown in Supplementary Figure 14d, e and f. For Figure S1e, the westerns have been added.

2. Inhibitor treatment period varies across the figures. (For example, in Figure 1 cells were treated for 8 hours. The same cell type in Figure 5 and Supplementary Fig2 were subjected to treatment for 18 or 16 hours.

Response: We can detect pY54-H3 by 8 hours of inhibitor treatment and that is why most immunoblotting and ChIP experiments were performed post 8 hours of treatment. However, to

validate NCOR binding to pY54-H3 marks, we wanted to build high levels of pY54-H3 mark, and that is why these experiments were performed after 16-18 hours post treatment.

3. What is the effect of inhibitor treatment on the total protein level of SHP2 and ACK1?

Response: Effect on total protein is shown in Figure 2b

4. Figure 1F: SHP2 activity in these tissues is important to be shown.

Response: pSHP2 westerns of ACK1 KO mice organs is now included in Figure 1f.

5. If AR is shuttling SHP2, it is not clear on why SHP2 activity is required for its nuclear translocation (Fig 2C).

Response: ACK1 forms a complex with AR in cytosol (our earlier *Cancer Cell* publication). ACK1 also phosphorylates SHP2 in cytosol, and the pSHP2 then interacts with AR and piggyback AR to nucleus (our model, Figure 8h). We believe that mutant SHP2 is not phosphorylated by ACK1 compromises its ability to form complex with AR, which reflects in its poor nuclear translocation.

6. In Figure 2B it will be better to run cytoplasmic and nuclear extracts in the same SDS PAGE gel to claim nucleocytoplasmic shuttling of pSHP2.

Response: Well, it is technically challenging for number of reasons; first, the number of lanes exceeds 10 (we get the best resolutions in 10-well gels), second, the amounts of proteins will be different in two cases so immunoblotting exposures can be challenging. In any case, our current data provides fairly clear picture of the phenomenon.

7. Supplementary Fig S2C: If Ack1 is upstream of SHP2 activity as per the hypothesis, why will SHP2 inhibition lead to reduction in Ack1 phosphorylation.

Response: ACK1 upon phosphorylation is known to be polyubiquitinated and degraded. We have seen that at high concentration of SHP099, pY-ACK1 levels are reduced, likely due to its enhanced phosphorylation dependent ubiquitination/degradation.

8. What is the difference between supplementary figures 4A and 4E?

Response: Thanks you so much for looking carefully. Yes, there was error in labelling in Figure 4E (it should be AR instead of ACK1). We fixed the error.

9. Supplementary Fig. S12A- figure legend says ChIP was performed using NCoR1 & NCoR2 (SMRT) antibodies but data shown only for NCoR1 in VCaP cells.

Response: Thank you for pointing out. We have corrected the figure legend.

10. pSHP-2 antibody detects which phosphorylation?

Response: We have used primarily Tyr580-SHP2 antibodies.

11. In Supplementary Fig 4D pY54-H3 signal is in the cytoplasm (Overexpression of SHP-2 279 mutant).

Response: We believe it was the overexposure (caused due to autoexposure) which caused the signal to be there. We went and checket it. The image obtained using the same exposure time for WT is now included for Y279. Once again thank you for looking so carefully.

12. Does SHP2 binds directly to chromatin? Or it remains bound to AR and regulates H3 phosphorylation?

Response: Have we done ChIP with SHP2 antibodies upon tratment of C4-2b cells with inhibitors, please see the data shown in Supplementary Figure 10f. *In vitro* assay indicates that

histones/nucleosomes were dephosphorylated by SHP2 (point#2). Taken together, these data suggests that SHP2 could binds directly to chromatin. Whether, while bound to chromatin, it is also bound to AR remains to be determined.

13. Supplementary data S3A-C: Mass spec data for all the identified proteins (not just SHP2) need to be shown in the manuscript as a list may be.

Response: A list of all the Tyr-phosphopeptides detected by mass spec is deposited in PRIDE database: Project accession: PXD037546.

14. Is Ack1 shown to directly phosphorylate Y279 and Y580 sites on SHP2?

Response: Jenkins et al. (*Science Signaling*, 2018) demonstrated SHP2 Y542- and Y580-phosphorylation by ACK1. Further, a significant increase in phosphorylated p44/42 MAPK was seen with the co-expression of SHP2 E76K and ACK1. Moreover, mutation of either Y542 or Y580 in SHP2 resulted in reduction of phospho-p44/42 MAPK to baseline levels. Together, their data suggested that SHP2 with the activating E76K mutant, which is thought to allow SHP2 to constitutively adopt the open, active conformation, requires phosphorylation of both Tyr542 and Tyr580 by ACK1 for full activation of downstream signaling.

15. The authors mention in the discussion that the direct chromatin-modifying role for protein tyrosine phosphatases (PTPs) is not known and this is the first report of the epigenetic activity among the PTP superfamily of enzymes. This may not be entirely true. The authors may check the literature carefully and modify this statement.

Response: We have edited the statement.

16. "PTPN2 (TCPTP), PTPN6 (SHP1) and PTPN22 (PEP) have strong presence in nuclei of immune cells" This statement is factually incorrect. None of the references cited for the same does support it.

Response: We have performed literature survey and nuclear expression of these proteins are described below. Please note that these are not necessarily in immune cells, so accordingly, we have edited sentence as follows (and provided following references): *'PTPN2 (TCPTP), PTPN6 (SHP1) and PTPN22 (PEP) have presence in the nuclei'*.

PTPN2 (TCPTP or TC45)

- a. ten Hoeve J., Ibarra-Sanchez M. J., Fu Y., Zhu W., Tremblay M., David M., Shuai K. Identification of a nuclear Stat1 protein tyrosine phosphatase. *Mol. Cell. Biol.* 2002;22:5662–5668.
- b. Yamamoto T., Sekine Y., Kashima K., Kubota A., Sato N., Aoki N., Matsuda T. The nuclear isoform of protein-tyrosine phosphatase TC-PTP regulates interleukin-6-mediated signaling pathway through STAT3 dephosphorylation. *Biochem. Biophys. Res. Commun.* 2002;297:811–817.
- c. Aoki N., Matsuda T. A nuclear protein tyrosine phosphatase TC-PTP is a potential negative regulator of the PRL-mediated signaling pathway: dephosphorylation and deactivation of signal transducer and activator of transcription 5a and 5b by TC-PTP in nucleus. *Mol. Endocrinol.* 2002;16:58–69.

PTPN6 (SHP1)

Craggs G., Kellie S. (2001) A functional nuclear localization sequence in the C-terminal domain of SHP-1. *J. Biol. Chem.* 276, 23719–23725.

PTPN22 (PEP)

a. Flores E, Roy G, Patel D, Shaw A, Thomas ML. 1994. Nuclear localization of the PEP protein tyrosine phosphatase. *Mol Cell Biol* 14: 4938–46

b. Gyorloff-Wingren A, Saxena M, Han S, Wang X, Alonso A, Renedo M, Oh P, Williams S, Schnitzer J, Mustelin T. 2000. Subcellular localization of intracellular protein tyrosine phosphatases in T cells. *Eur J Immunol* 30: 2412–21

17. Most of the western blot labelling is confusing and some figures have labelling mistakes.

Response: Thanks. We have rectified the mistakes.

Reviewer #3 (Remarks to the Author):

This manuscript by Chouhan et al, describes a novel role for the protein tyrosine phosphatase Shp2 in epigenetic regulation. The authors provide intriguing evidence to support the model that Shp2 through the actions of the kinase, ACK1/TNK2 is trafficked to the nucleus via the androgen receptor (AR) where it “erases” tyrosyl phosphorylated Histone3 at residue 54. The strength of the model is on the novel relationship between Shp2 and the effects of modulation pY54-H3 marks as an epigenetic signal. However, the paper is constructed in a confusing manner because it jumps between multiple cell lines, the Noonan syndrome with multiple lentigines (NSML) model and prostate cancer models to substantiate the clinical importance of the findings. There are additional mechanistic shortcomings in the manuscript that need to be addressed. Most notably more convincing evidence that pY54-H3 is a bona fide Shp2 substrate is required if indeed that is the stated mechanism that is being put forth. Other interpretations of the data are conceivable. Overall, this is a very interesting study however, in its current form additional experiments are required to directly support the proposed model.

Major Comments

1. The authors should provide some rationale for the selection of Shp2 for its ability to modulate H3 tyrosyl phosphorylation. There are other nuclear resident PTPs (e.g. TC-PTP) that would likely serve as equally relevant to test in this model. It seems out of the blue as to how they arrived at Shp2 – what about Shp1?

Response: Yes, we have assessed other nuclear phosphatases including TC-PTP and SHP1. The data is enclosed, please see Supplementary 14 e, f. As we have responded to referee #2, indeed other phosphatases could potentially regulate pY54-H3 levels, possibly in other cell types.

2. It is unclear how Shp2 is regulated in this model. Shp2 exists in a closed conformation and in this state the basal activity of the phosphatase is typically less than 5% of its maximal. In order for Shp2 to be activated engagement of the SH2 domains are required in all instances. Phosphorylation of Shp2 has only minor modulatory effects on overall catalytic activity and this is largely consistent with the downstream signaling effects. The authors refer to the Jenkins et al work and state that “ACK1 phosphorylates SHP2 leading to activation of its Tyr phosphatase activity”. Unfortunately, Shp2 tyrosyl phosphorylation at either Y542 or Y580 as it appears to have been used here is not a proxy (and at least not a reliable one) for intrinsic Shp2 catalysis.

Response: Firstly, we have edited the sentence “*ACK1 phosphorylates SHP2 leading to activation of its Tyr phosphatase activity*” to “*ACK1 regulates SHP2 mediated H3 Tyr54-dephosphorylation activity*”, since our data is consistent with this statement. ACK1 kinase upon overexpression, or activation by other RTKs, is known to autoactivate itself, which is seen in form of a robust auto-phosphorylation at Tyr284 site (our earlier publications). In addition, at least three other sites, Tyr518, Tyr827 and Tyr859, are also phosphorylated in ACK1. We believe pTyr284 and pTyr518/pTyr827 sites might be recognized by two SH2 domains of SHP2 leading to its open conformation and its enhanced catalytic/phosphatase activity. As a consequence, ACK1 could now phosphorylate SHP2 at Tyr580.

C-terminal SH3 domains of ACK1 has ability to interact with proline-rich regions as well as play a crucial role in substrate recognition. It opens up a possibility that ACK1 SH3 domain may be involved in recognizing the SHP2 region with Y580 and Y542 sites. Thus, 580 mutant of SHP2 may not be optimally recognized by ACK1 and thereby compromising its phosphatase activity.

3. The authors need to substantiate that the effects on pY54-H3 are a direct consequence of Shp2 phosphatase acting on this substrate.

Response: We have addressed this concern, described above as a response to Referee#2. Briefly, purified SHP2 was incubated with the H3 derived peptide (spanning 49-63 aa) with Tyr54 phosphorylated. SHP2 erased Y54-phosphorylation (**Supplementary Fig. S1f**). In addition, purified nucleosomes were incubated with purified ACK1 and phosphorylated nucleosomes were then incubated with purified SHP2. ACK1 enhanced pY54-H3 in nucleosomes (**Supplementary Fig. S1g**, lane 2), which was significantly compromised upon SHP2 incubation (**Supplementary Fig. S1g**, lane 4). Overall these data indicates direct dephosphorylation of pY54-H3 by SHP2.

4. What is the effects of pY54-H3 levels when Shp2 is subjected to an in vitro dephosphorylation assay? The authors should use their pY54-H3 (44-59): VALREIRRpYQKSTE peptide as a substrate for in vitro dephosphorylation assays.

Response: Yes we performed this experiment, please see above.

5. Can the authors use substrate-trapping mutants of Shp2 to show direct enzyme-substrate complex formation?

Response: Substrate-trapping experiment is shown, please see **Supplementary Figure 1i**.

6. If Shp2 needs to go to the nucleus as a direct mechanism of pY54-H3 dephosphorylation then does abrogating Shp2/AR complex prevent the dephosphorylation of pY54-H3.

Response: Correct. AR antagonists, Enzalutamide and Abiraterone treatment resulted in enhanced pY54-H3 levels (**Figure 2a**). Similarly, AR-Y267F, -Y307F, -Y534F and Y223/363F mutants bound poorly to SHP2, consequently causing increase in pY54-H3 levels (**Supplementary Fig. 10a and b**), suggesting that failure of mutant-AR/SHP2 complex formation compromise SHP2 translocation into nucleus.

7. The authors cannot conclude as stated in line 175-176 that ACK1 mediated Shp2 phosphorylation is crucial to acquire its optimal phosphatase activity.

Response: As we mentioned above, we have edited the sentence

8. What is the explanation in Figure 2 for why AR and Shp2 antagonism are required for their interaction.

Why do all the inhibitors result in nearly a complete inhibition in the expression levels of AR (see Fig. 2A, panel 5)? If this is correct, it might suggest that the reduced complex formation between Shp2 and Ack is mediated by AR.

Response: ACK1 is in complex with AR (our earlier *Cancer Cell* publication). Essentially, ACK1 kinase inhibition (by R-9b) or AR antagonism by Enzalutamide or Abiraterone could disrupt this ACK1/AR complex, which will be also reflected in loss of ACK1/SHP2 complex. This also explains why ACK1 & SHP2 inhibitors as well as AR antagonists compromises the AR transcription levels (seen as loss of AR proteins in panel 5)

10. What is the distribution between the nuclear and cytosolic pools of total Shp2 this needs to be included in the immunoblot as a control and proper quantitation of these levels performed.

Response: We have added this data, please see Figure 2b.

11. It is not clear what is meant in line 206 by “nuclear depletion of Shp2” – is this what the authors really mean.

Response: We have edited this sentence.

12. An important control is missing, in Figure 8A, and that is the expression levels of AR in prostate, testis and liver.

Response: The AR levels are shown.

13. What are the effects of the pY54-H3 status with the Noonan syndrome mutants (e.g. N308D, D61G).

Response: The data is shown, please see Supplementary Figure 13d.

Minor comments:

1. Why is that the authors refer to the actions of Shp2 as an ‘eraser’ whilst it is understandable that this phrase often used in the epigenetic field, strictly speaking it is jargon. The authors should refer to the actions of the Shp2 on pY54 as dephosphorylation rather than the more ambiguous term of an “eraser”. This term is suitable for the conclusion but in the description of the results they need to be clearer.

Response: We deleted the word eraser.

2. The use of multiple cell lines is often not well justified and made it difficult to understand what the rationale for a particular cell line was being used for (eg LNCaP, C4-2B, VCap, LAPC-4 etc). This makes it hard to ascribe consistency with the model.

Response: Yes, we understand the concern, however, there are reasons why we need to use these 4 cell lines. All of these four are AR, ACK1 and SHP2 positive, and responds well to (R)-9b and SHP99 treatments, which was essential for us to validate the phenomenon. Further, these are well established and commonly used prostate cancer cell lines capable of forming tumors in mice. We have reason to believe that AR positive breast cancer cell lines (ACK1 has robust expression in breast cancers) would also exhibit this phenomenon.

3. In the figures the cell type is indicated and in others it is not, can the authors be consistent and show the cell type given that many are used in this study.

Response: In figures where more than one cell line was used, cell lines were mentioned. In figures where just one cell lines was used, cell line was mentioned in Figure legend.

4. Figure legends should state the statistical analyses used.

Response: It is done.

5. The model presented in Figure 9 is a little unclear and should be more precisely generated. For example, it shows the N-SH2 domain of Shp2 binding to phosphorylated (pY) AR. This data was neither demonstrated nor referenced in this study. Why is the AR missing in the NSML patient model?

Response: Agree, we have no direct evidence of N-SH2 domain of SHP2 binding to pY-AR, however, AR-Y267F, -Y307F, -Y534F and Y223/363F mutants bound poorly to SHP2, consequently causing increase in pY54-H3 levels (**Supplementary Fig. 10a and b**), suggesting that pY-AR forms complex with SHP2. Since pY is primarily recognized by SH2 domains, two of which are present in SHP2, it was suggested to be plausible domain of interaction, however we have made changes in the model reflecting this discussion.

Reviewers' comments:

Reviewer #1 (Remarks to the Author):

The authors addressed all issues raised in a professional and thorough manner. They added experimental data and extended the discussion on their work and its significance to the field. I strongly support publication of the work.

Reviewer #2 (Remarks to the Author):

Chouhan et al has included some new experiments to address the reviewer's concerns in the revised version of manuscript. However, concerns still exist with the specificity of SHP2 for dephosphorylation of pTyr54-H3. As per the comments on the previous version of the manuscript, the specificity was a major concern. And the authors do show in the revised version that multiple phosphatases (including PTPN1 (PTP1B), PTPN2 (TCPTP), PTPN6 (SHP1) and PTPN11) are all able to dephosphorylate this residue. Given so much promiscuity, it is hard to gauge the importance of only SHP2-H3 connection in this process.

Also, another concern is if SHP2 is a direct phosphatase for H3. The authors performed two supplementary experiments using H3 peptides. But, it is surprising that no controls were included in these experiments by the authors. The enzyme was purified from cells. How do you ascertain that SHP2 is the dephosphorylating enzyme but not any associated/interacting protein (phosphatase) that came in the purification is dephosphorylating H3. With lack of appropriate specificity controls, it is not entirely convincing that SHP2 dephosphorylates H3.

Reviewer #3 (Remarks to the Author):

The authors have provided explanation to most of the concerns raised in the initial review. However, there are issues that still remain from concerns raised in the initial review.

The most concerning issue is the one that addresses how Shp2 is activated in the nucleus. The authors do not provide any additional data/experiments to address this question, nor do they provide a satisfactory explanation as to how Shp2 gets activated in the nucleus. They speculate that "pTyr284 and pTyr518/pTyr827 sites might be recognized by two SH2 domains of SHP2 leading to its open conformation and its enhanced catalytic/phosphatase activity." Inspection of these sites make this unlikely since neither pY284, pY518 nor pY827 fall into the consensus binding site that is recognized by the SH2 domains of Shp2. If the authors cannot plausibly explain how Shp2 is activated it makes the issue of it acting to dephosphorylate pY54-H3 very hard to reconcile. This in context of the relatively minor role for pY542/580 for Shp2 activation make this component of the model not particularly compelling. Which in combination with the point below is concerning.

The other area of concern that continues to be problematic is the evidence demonstrating that Shp2 directly dephosphorylates pY54-H3 in particular within a cellular context. The authors present in Supplementary Figure 1b – panel i the substrate-trapping experiments with Shp2 and H3. However, here they show that the substrate-trapping mutants (D425A/C459S and C459S) all bind to H3 to levels that are equivalent to that of wild-type Shp2. Therefore, the interaction is independent of pY54-H3 as one would have expected increased H3 interactions with the substrate-trapping mutants. The authors do not address this, and it raises concerns about whether H3 is dephosphorylated by Shp2 in cells.

Reviewer #1 (Remarks to the Author):

The authors addressed all issues raised in a professional and thorough manner. They added experimental data and extended the discussion on their work and its significance to the field. I strongly support publication of the work.

Response: Thank you so much.

Reviewer #2 (Remarks to the Author):

1. Chouhan et al has included some new experiments to address the reviewer's concerns in the revised version of manuscript. However, concerns still exist with the specificity of SHP2 for dephosphorylation of pTyr54-H3. As per the comments on the previous version of the manuscript, the specificity was a major concern. And the authors do show in the revised version that multiple phosphatases (including PTPN1, PTPN2, PTPN6 and PTPN11) are all able to dephosphorylate this residue. Given so much promiscuity, it is hard to gauge the importance of only SHP2-H3 connection in this process.

Response: This manuscript reports the discovery of pY54-H3 epigenetic mark and provides detail characterization of this event that includes, its importance in genetic disorder such as NSML, as well as in prostate cancer. Further, we show that ACK1 kinase mediated SHP2 phosphatase activation is critical for modulation of pY54-H3 epigenetic mark. These data open up an entirely new topic in the field of phosphatase, the epigenetic regulation by Tyr-phosphatases. Extensive dataset that we have provided in this manuscript leaves no doubt that this discovery is highly relevant and important for cancer therapeutics. In addition, it provides possible treatment modality for genetic disorders such as NSML, for which no options exist.

As a **purely discussion point**, we considered the '**possibility**' of other phosphatases to act on pY54-H3. The rationale was explained in our earlier point-by-point response; here we provide further evidence/explanation. Briefly, multiple Tyrosine phosphatases are known to target same substrates; please refer to a publication by Benjamin Neel and Igor Stagljjar, '*A Global Analysis of the Receptor Tyrosine Kinase-Protein Phosphatase Interactome*' that shows multiple such examples, including FGFR4 and KDR targeted by PTPN6, PTPN11 and PTN12.

[https://www.cell.com/molecular-cell/pdfExtended/S1097-2765\(16\)30817-6](https://www.cell.com/molecular-cell/pdfExtended/S1097-2765(16)30817-6)

Epigenetic marks are also known to be targeted/erased by multiple enzymes. Since this is the first report of erasing histone phosphorylation, we have provided examples of another epigenetic mark, erasing of histone methylation.

As shown above, each of the histone H3-K4, H3-K9, H3-K27, H3-K36 and H4-20 methylation marks are erased/targeted by multiple histone demethylases. It does not affect specificity of the interaction, as those have distinct functional relevance.

It is to be noted that this manuscript describes pY54-H3 regulation by SHP2/PTPN11, which we saw primarily in male reproductive system. The functional relevance of pY54-H3 regulation by PTPN1, PTPN2, PTPN6 may be relevant in other tissue type, or in different stage of development. These are the future studies, which will provide broad functional relevance for pY54-H3 epigenetic marks.

Overall, respectfully, we do not believe the ability of other phosphatases to target pY54-H3 (assuming such activity exist in hitherto unknown tissue type) anyway affects specificity of SHP2's ability to act on pY54-H3. This manuscript provides robust data to indicate (i) presence of novel pY54-H3 epigenetic marks, (ii) which are regulated by SHP2, and (iii) have crucial role in androgen receptor regulation.

Since Referee#1 and #3 did not have any issues, we retained this brief discussion (third last paragraph in Discussion section), however, if Referee#2 still have objection, we are open to idea of deleting this brief discussion on other phosphatase potentially targeting pY54-H3 epigenetic marks.

2. Also, another concern is if SHP2 is a direct phosphatase for H3. The authors performed two supplementary experiments using H3 peptides. But, it is surprising that no controls were included in these experiments by the authors. The enzyme was purified from cells. How do you ascertain that SHP2 is the dephosphorylating enzyme but not any associated/interacting protein (phosphatase) that came in the purification is dephosphorylating H3. With lack of appropriate specificity controls, it is not entirely convincing that SHP2 dephosphorylates H3.

Response: The recombinant FLAG-tagged SHP2 protein used in this study was purified using FLAG beads, followed by elution. The purity of the SHP2 was confirmed by Coomassie blue staining (and the western blot). Please see **Supplementary Figure 1g** and **1h**. As shown on right, this process yielded highly purified SHP2 protein; we believe this preparation does not contain any other phosphatases as an interacting protein/s.

This purified SHP2 protein was then used for experiments using peptides, which includes many controls. Please see **Supplementary Figure 1i** to **1m**. As shown below, purified SHP2 targeted H3 peptide containing phosphorylated Y54 (i). As a control, we used peptides corresponding to two other known histone phosphorylations, pTyr37 in H2B, and pTyr88 in H4. The phospho-peptides corresponding to these two modified residues were untouched by purified SHP2. Further, ac130-H2A and pY363-AR peptide were used as additional controls, which were also spared by SHP2.

Overall these data clearly establishes SHP2 as a direct phosphatase for H3 and does not dephosphorylates other histone phosphorylations.

Reviewer #3 (Remarks to the Author):

The authors have provided explanation to most of the concerns raised in the initial review. However, there are issues that still remain from concerns raised in the initial review.

Response: Thanks. We have addressed remaining concerns below.

1. The most concerning issue is the one that addresses how Shp2 is activated in the nucleus. The authors do not provide any additional data/experiments to address this question, nor do they provide a satisfactory explanation as to how Shp2 gets activated in the nucleus. They speculate that “pTyr284 and pTyr518/pTyr827 sites might be recognized by two SH2 domains of SHP2 leading to its open conformation and its enhanced catalytic/phosphatase activity.” Inspection of these sites make this unlikely since neither pY284, pY518 nor pY827 fall into the consensus binding site that is recognized by the SH2 domains of Shp2. If the authors cannot plausibly explain how Shp2 is activated it makes the issue of it acting to dephosphorylate pY54-H3 very hard to reconcile. This in context of the relatively minor role for pY542/580 for Shp2 activation make this component of the model not particularly compelling. Which in combination with the point below is concerning.

Response: As per Diop *et al.* the two SH2 domains of SHP2 (N and C terminal) makes contact with Tyr-phosphorylated residues based on pY-Ψ-x-Ψ motif (Ψ being hydrophobic residue). <https://www.ncbi.nlm.nih.gov/pmc/articles/PMC9783222/>

We carefully checked all of the known Tyr-phosphorylation events in ACK1 and observed that potentially ACK1 pTyr518 (pYDPV) and ACK1 pTyr859/860 (YpYLLP) could make contacts with two SH2 domains of SHP2.

Our crystallographer collaborators (we have solved crystal structure of ACK1 kinase domain recently) then took a careful look at the ACK1/SHP2 interaction. Briefly, both SH2 domains

in SHP2 (PDB 2SHP) are accessible for phosphoTyr-peptides. We docked a phospho-Tyr (PDB 1KC2) onto each of the SHP2 SH2 domains and observed that the phosphopeptides fit. Overall, ACK1 pY518 and pY859/Y860 fit well with two SH2 domain of SHP2. Please see the data shown below. We have also added this data as **Supplementary Figure 14d-g**.

To summarize:

- The respective peptides from ACK1 would fit into the two SH2 domains of SHP2
- Significantly, the sequence between Y518 and Y859/Y860 is Proline-rich, which provides enough flexibility, thus potentially allowing ACK1 Y518 and Y859/Y860 to bind to both SH2 domains simultaneously (see below in model).

Based on these data, we have refined our model of ACK1 interaction with SHP2 wherein two phosphorylation sites pY859/Y860 and pY518 interacts with two SH2 domains of SHP2, bringing ACK1 kinase domain in vicinity of SHP2 phosphatase domain, facilitating its activation. The revised graphical model is presented on right (also shown in revised manuscript).

2. The other area of concern that continues to be problematic is the evidence demonstrating that Shp2 directly dephosphorylates pY54-H3 in particular within a cellular context. The authors present in Supplementary Figure 1b – panel i the substrate-trapping experiments with Shp2 and H3. However, here they show that the substrate-trapping mutants (D425A/C459S and C459S) all bind to H3 to levels that are equivalent to that of wild-type Shp2. Therefore, the interaction is independent of pY54-H3 as one would have expected increased H3 interactions with the substrate-trapping mutants. The authors do not address this, and it raises concerns about whether H3 is dephosphorylated by Shp2 in cells.

Response: We agree. The substrate tapping experiment need to be performed with pY54-H3 antibodies, rather than H3 antibodies. We performed the experiment, shown on right, and noticed that there was a significant increase in interaction of pY54-H3 with the substrate-trapping mutant (lane 3). This data has also been added in the revised manuscript (**Supplementary Figure 1o**).

Overall, we believe, we have addressed all the concerns of reviewers.

REVIEWERS' COMMENTS

Reviewer #3 (Remarks to the Author):

I am satisfied with the response of the authors and all concerns have been addressed.